# TRAINING-FREE LLM VERIFICATION VIA RECYCLING FEW-SHOT EXAMPLES

## ABSTRACT

Although LLMs have achieved remarkable performance, the inherent stochasticity of their reasoning process and varying conclusions present significant challenges. Majority voting or Best-of-N with external verification models has been explored to find the most promising solution among multiple LLM outputs. However, these approaches have certain limitations, such as limited applicability or the cost of an additional training step. To address this problem, we propose a novel and effective framework that **Re**cycles **Fe**w-shot examples to ve**ri**fy LLM outputs (ReFeri). Our key idea is to additionally utilize the given few-shot examples to evaluate the candidate outputs of the target query, not only using them to generate outputs as the conventional few-shot prompting setup. Specifically, ReFeri evaluates the generated outputs by combining two different scores, designed motivated by Bayes' rule, and subsequently selects the candidate that is both confidently determined and contextually coherent through a few additional LLM inferences. Experiments with three different LLMs and across seven diverse tasks demonstrate that our framework significantly improves the accuracy of LLM–achieving an average gain of 4.5%–through effective response selection, without additional training.

## 1 INTRODUCTION

Recently, large language models (LLMs) have shown remarkable performance in many real-world tasks involving complex reasoning, such as math, coding, and robotics (Anthropic, 2024; Dubey et al., 2024; OpenAI, 2024c; Team et al., 2023). To enhance the reasoning capacity of LLMs, various approaches have been proposed, ranging from in-context learning at test time (Wei et al., 2022; Kojima et al., 2022) to recent RL training method (Qu et al., 2024; Guo et al., 2025). Despite these improvements, the inherent stochastic nature of LLM still presents significant challenges, since different reasoning paths can be generated for the same input and can lead to varying conclusions (Kadavath et al., 2022; Wang & Zhou, 2024; Qiu & Miikkulainen, 2024). Majority voting approaches, such as self-consistency (Wang et al., 2023b; Aggarwal et al., 2023), have been widely adopted to reduce such randomness by aggregating multiple LLM outputs and determining a single prediction. However, this approach is only applicable when the answer can be easily extracted from the output and aggregated. Consequently, it is difficult to apply to open-ended text generation tasks such as summarization and personalized chatbot (Stiennon et al., 2020; Salemi et al., 2024).

To address this challenge, finding the most promising one among multiple LLM outputs using a specific selection method, often called *Best-of-N*, has recently gained attention (Snell et al., 2024; Gui et al., 2024). For instance, one of the most representative approaches is to score each output using external verification models such as Outcome Reward Models (ORMs) (Cobbe et al., 2021; Uesato et al., 2022) or Process Reward Models (PRMs) (Lightman et al., 2024; Wang et al., 2024b), and then selecting the highest-scoring output. However, to obtain these reward models, training with a large amount of task-specific labeled data is often necessary; therefore, applying this framework to specific target domain, which is far from well-explored domains such as math and coding, is challenging. Prompting LLM to select the most promising output–such as *LLM-as-judge*–is considerable to remove the reliance on the verification model (Chen et al., 2023; Zheng et al., 2023). However, this approach is only effective when the given LLM has sufficient intrinsic knowledge for the target domain; consequently, it often requires separate training steps and datasets again to achieve sufficient performance (Yuan et al., 2024; Mahan et al., 2024; Zhang et al., 2025).

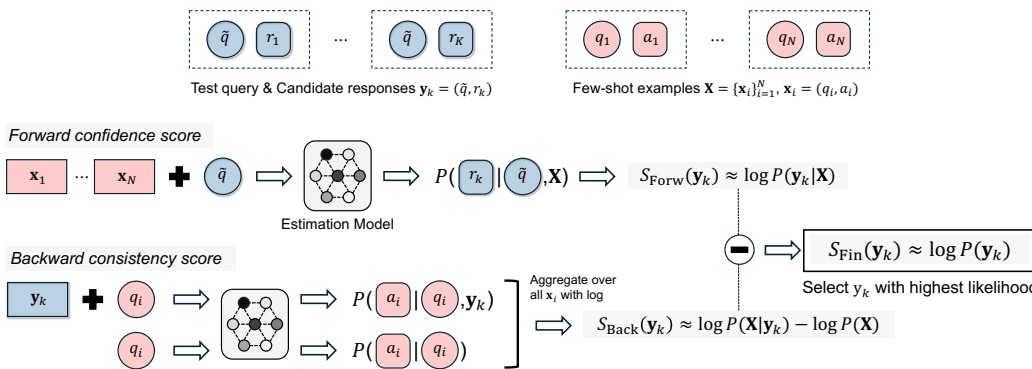

Figure 1: **An overview of ReFeri.** For $K$ candidate responses from LLMs, ReFeri assigns each candidate a forward confidence score (*how likely candidate is to be generated conditioned on few-shot examples*) and a backward consistency score (*how well candidate explains the answers of few-shot examples*). Then, the response with the best joint score is selected as the final answer.

Motivated by this, we suggest a new perspective: *utilization of few-shot examples to verify and select among multiple LLM outputs*. As recent LLMs have been trained with an extensive instruction tuning step, they often exhibit better performance without few-shot examples (Guo et al., 2025; Sprague et al., 2025), and hence using these examples at test time is recently losing attention (see results in Table 1). However, we argue that using few-shot examples is still one of the easiest and most direct ways to let LLMs know how to solve the given task with human prior knowledge, even if LLMs have not encountered it before. Therefore, in this work, we provide a new framework that enables better exploitation of few-shot examples by using them not only for generating multiple outputs, but also for selecting the most promising one.

**Contribution.** In this work, we propose **ReFeri**, a novel and effective framework that **Re**cycles **Fe**w-shot examples to ve**ri**fy LLM outputs. The core idea of ReFeri is additionally utilizing the given few-shot examples to evaluate the candidate outputs of the target query, not only using them to generate outputs as conventional few-shot in-context learning.[1] Specifically, ReFeri estimates the likelihood of the generated outputs by decomposing it into two different scores (*forward confidence score* and *backward consistency score*) conditioned on few-shot examples, which are derived from *Bayes' rule*. The forward confidence score measures the likelihood of candidate outputs given the few-shot examples and the test query, favoring more confident ones. On the other hand, the backward consistency score measures whether conditioning on the candidate output well explains the likelihood of the few-shot examples compared to conditioning on their queries alone. By combining these scores, ReFeri selects the candidate that is both confidently determined and contextually coherent through a few additional LLM inferences. Consequently, ReFeri does not require additional model training to select the most promising output, and allows better leverage of both intrinsic knowledge of LLM and human prior within the provided few-shot examples. See Figure 1 for the illustration.

We validate the effectiveness of ReFeri across three different LLMs (GPT-4o, GPT-4o-mini, and LLaMA-3.1-8B) and seven different benchmarks. When selecting one response among five candidates generated by few-shot chain-of-thought (CoT) prompting, ReFeri consistently outperforms other training-free selection across all tasks, with an average gain of 4.5% over random selection and 2.4% over prompt-based selection methods (see Figure 2). ReFeri also scales reliably with the number of candidate responses, demonstrating its practical utility in test-time scaling. To better understand the behavior of ReFeri, we conduct more complementary analyses, showing that our method is robust to variations in few-shot example selection, prompt template choices, and the choice of model used for likelihood estimation; ReFeri yields consistent improvements without reliance on specific prompt templates or few-shot ex-

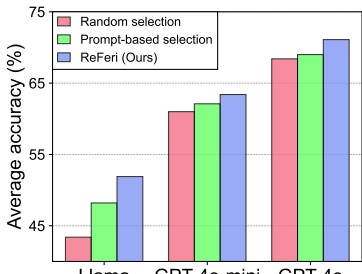

Figure 2: **Summary of results.** Average accuracy across seven benchmarks with training-free selection methods. ReFeri shows consistent effectiveness (see Section 3.2).

---

[1]In-context learning uses given few-shot examples as additional input context upon the target query.

amples. Moreover, when combined with smaller likelihood estimators, ReFeri significantly reduces the cost per query compared to larger baseline models while outperforming them.

## 2 TRAINING-FREE LLM VERIFICATION VIA RECYCLING FEW-SHOT DATA

### 2.1 PROBLEM FORMULATION

Let us denote LLM as $\mathcal{M}$ and a given test query as $\widetilde{q}$. We assume that we have $N$-shot examples $\mathbf{X} = \{\mathbf{x}_i\}_{i=1}^N, \mathbf{x}_i = (q_i, a_i)$ where $q_i$ is another input query from the same task and $a_i$ is the ground-truth answer, which can be provided by human annotator or generated by LLM itself. Then, *few-shot prompting* incorporates the few-shot examples $\mathbf{x}_i$ in $\mathbf{X}$ as additional input context to obtain the response $r_k$, which is expected to be improved thanks to the in-context learning capability of LLMs:

$$r_k \sim \mathcal{M}(\widetilde{q}, \mathbf{X}), \tag{1}$$

where multiple non-identical predictions $r_k, k = 1, \ldots, K$ can be sampled. Then, our goal is to find the most appropriate response $r_{k^*}$ among them. For example, the self-consistency method (Wang et al., 2023b) simply applies majority voting to determine the single prediction. On the other hand, the best-of-K method uses the external verifier such as reward models (Cobbe et al., 2021; Lightman et al., 2024) to score the predictions and select the highest scored one. Formally, with the external verifier $R_\phi$, it can be described as below:

$$r_{k^*} = \arg \max_{k=1,\ldots,K} R_\phi(\mathbf{y}_k), \tag{2}$$

where $\mathbf{y}_k = (\widetilde{q}, r_k)$. While these approaches are widely used in practice, there are certain challenges due to the limited applicability and the need for a verification model for the target task.

### 2.2 REFERI: VERIFYING LLM OUTPUTS WITH BAYES' RULE WITH FEW-SHOT DATA

In this section, we introduce a framework that selects candidates from LLM by **Re**cycling **Fe**w-shot examples for the ve**ri**fication (**ReFeri**). The core idea of ReFeri is to leverage few-shot examples not only for generation but also for validation, thereby recycling them to score and select answers without additional training. Specifically, ReFeri estimates the plausibility of each answer candidate by combining two complementary signals: (1) a *forward confidence score* which captures how likely the model is to generate response $r_k$ given test query $\widetilde{q}$, few-shot examples $\mathbf{X}$, and (2) a *backward consistency score*, measuring how $r_k$ is effective to correctly answer the queries $q_i$ in $\mathbf{X}$.

**Problem setup.** Let us assume that we have an estimation model $P$ which can measure the likelihood $P(\mathbf{y}_k) = P(r_k \mid \widetilde{q})$ of the response $r_k$ conditioned on the given query $\widetilde{q}$.[2] Then, our goal is to select the response $r_{k^*}$ which yields the highest likelihood if the estimation is accurate:

$$k^* = \arg \max_{k=1,\ldots,K} P(\mathbf{y}_k). \tag{3}$$

We note that the likelihood has shown effectiveness to find high-quality reasoning path (Wang & Zhou, 2024). However, selecting based on the estimated $P(\mathbf{y}_k)$ could be ineffective in practice, as it entirely depends on the estimation model's intrinsic knowledge, which can be limited in unfamiliar or challenging domains. Furthermore, when there is a mismatch between $\mathcal{M}$ and $P$, the estimated likelihoods can be unreliable as minor syntactic variations in response can make large deviations. To address this, we propose to reinterpret $P(\mathbf{y_k})$ with few-shot examples $\mathbf{X}$, through Bayes' rule:

$$P(\mathbf{y}_k) = \frac{P(\mathbf{y}_k \mid \mathbf{X}) \cdot P(\mathbf{X})}{P(\mathbf{X} \mid \mathbf{y}_k)}. \tag{4}$$

Then, in the log form, this can be decomposed into two intuitive forward and backward scores:

$$\log P(\mathbf{y}_k) = \underbrace{\log P(\mathbf{y}_k \mid \mathbf{X})}_{\text{forward}} - \underbrace{\left( \log P(\mathbf{X} \mid \mathbf{y}_k) - \log P(\mathbf{X}) \right)}_{\text{backward}}. \tag{5}$$

While Eq. 5 holds mathematically, discrepancies between the left- and right-hand sides can arise in practice due to the limitations of the estimation model. To address this, the core idea of ReFeri is to

---

[2]For the experiments in Section 3, we use pre-trained LLM as the estimation model.

estimate the forward and backward scores separately, as each can be more accurately approximated by the estimation model with the help of few-shot examples. Then, ReFeri combines these two estimated scores to yield the final selection score. Overall algorithm is presented in Algorithm 1.

**Forward confidence score.** Intuitively, $\log P(\mathbf{y}_k|\mathbf{X})$ captures the confidence of candidate response $r_k$ to test query $\widetilde{q}$; this score is high when $r_k$ well-aligns with the reasoning patterns in the few-shot examples $\mathbf{X}$. This forward score has certain advantages over direct estimation of $P(\mathbf{y}_k)$, as it allows the estimation to be grounded in the few-shot examples and hence reduces the reliance on its prior knowledge alone. As a result, the forward score provides a more context-aware and robust estimation, especially important in unfamiliar or domain-shifted scenarios. When the estimation model $P$ is equal to generation LLM $\mathcal{M}$, the forward score can be freely obtained during generation of $r_k$. Formally, under the autoregressive assumption for estimation model $P$, the forward score is derived as below:

$$S_{\texttt{Forw}}(\mathbf{y_k}) := \log P(\mathbf{y_k} \mid \mathbf{X}) = \frac{1}{T} \sum_{t=1}^{T} \log P(r_{k,t} \mid \widetilde{q}, \mathbf{X}, r_{k,<t}), \qquad (6)$$

where each candidate response is a sequence of $T$ tokens $r_k = (r_{k,1}, \ldots, r_{k,T})$. To avoid length bias, we apply a length normalized log probability (i.e., mean over $T$ tokens). Since ReFeri uses a separate estimation model, using raw logits without temperature scaling ($T = 1$) to ensure that the evaluation remains completely hyperparameter-free.

**Backward consistency score.** The backward score, $\log P(\mathbf{X}|\mathbf{y}_k) - \log P(\mathbf{X})$, evaluates how well the test query $\widetilde{q}$ and candidate response $r_k$ explains the few-shot examples $\mathbf{X}$. At a high level, this score serves as a form of consistency check between the response and the given few-shot examples. Under the assumption of mutual independence between few-shot examples, the backward score can also be derived similar to Eq. 6. To better utilize given few-shot examples, we refine the backward term using a leave-one-out strategy (Perez et al., 2021; Izacard et al., 2023) through prompt replacement; namely, we construct new demonstration $\widetilde{\mathbf{X}}_i$ by replacing $i$-th example $\mathbf{x}_i = (q_i, a_i)$ with a pair of test query and candidate response $(\widetilde{q}, r_k)$:

$$\widetilde{\mathbf{X}}_i := \mathbf{X}_{-i} \cup \{(\widetilde{q}, r_k)\}, \qquad (7)$$

where $\mathbf{X}_{-i}$ denotes the few-shot examples excluding $\mathbf{x}_i$. Then, by including $\widetilde{\mathbf{X}}_i$ during the estimation for $\mathbf{x}_i$ as additional input context similar to forward term, we define the modified backward score:

$$S_{\texttt{Back}}(\mathbf{y_k}) := \log P(\mathbf{X} \mid \mathbf{y}_k) - \log P(\mathbf{X}) = \frac{1}{N} \sum_{i=1}^{N} \big( \log P(a_i \mid q_i, \widetilde{\mathbf{X}}_i) - \log P(a_i \mid q_i) \big). \quad (8)$$

This inclusion of remaining examples $\mathbf{X}_{-i}$ enables more accurate estimation of the likelihood of target example $\mathbf{x}_i$ by leveraging the in-context learning capability of $P$ (see more discussions in Appendix B.9). Similar to Eq. 6, $\log P(a_i|q_i, \widetilde{\mathbf{X}}_i)$ and $\log P(a_i|q_i)$ can be calculated through a token-level decomposition using the autoregressive nature of $P$.

While the backward consistency score provides a reliable consistency signal, one may concern its computational cost as it grows linearly with the number of few-shot examples. To alleviate this, we propose a lightweight approximation; instead of iterating over all $N$ few-shot examples, we select only the single most relevant example to the test query $\widetilde{q}$. Specifically, we employ a pre-trained embedding model $E$ to encode both $\widetilde{q}$ and each $q_i$, and identify the most relevant example $\mathbf{x}_{i\dagger}$:

$$i^\dagger = \arg \max_{i=1,\ldots,N} \cos \big( E(\widetilde{q}), E(q_i) \big). \qquad (9)$$

Then, we define the approximated backward score $\widetilde{S}_{\texttt{Back}}$:

$$\widetilde{S}_{\texttt{Back}}(\mathbf{y_k}) := \log P(a_{i\dagger} \mid q_{i\dagger}, \widetilde{\mathbf{X}}_{i\dagger}) - \log P(a_{i\dagger} \mid q_{i\dagger}) \qquad (10)$$

**Final score.** By combining forward and backward scores following Eq. 5, we design our main selection score $S_{\texttt{Fin}}$ to find the most promising output $r_{k^\star}$ as below:

$$k^\star = \arg \max_{k=1,\ldots,K} S_{\texttt{Fin}}(\mathbf{y_k}), \quad S_{\texttt{Fin}}(\mathbf{y_k}) := S_{\texttt{Forw}}(\mathbf{y_k}) - \widetilde{S}_{\texttt{Back}}(\mathbf{y_k}). \qquad (11)$$

## 3 EXPERIMENTS

In this section, we design our experiments to investigate the following questions:

- ○ Is ReFeri effective to select the correct output across various tasks and LLMs? (Table 1)
- ○ Can ReFeri enable test-time scaling without external reward model and training? (Figure 3)
- ○ What is the effect of each component, and how robust is ReFeri? (Tables 2, 3, 4)
- ○ How does the estimation model affect cost and performance of ReFeri? (Figure 4, Table 17)

### 3.1 SETUPS

**Datasets.** We evaluate our method on seven benchmarks encompassing diverse reasoning paradigms, including symbolic-numeric, expertise-based, and multi-hop textual reasoning tasks. (1) *MATH500* (Lightman et al., 2024); a 500-problem subset of MATH (Hendrycks et al., 2021b), focused on symbolic manipulation and multi-step mathematical reasoning. (2) *MMLU-pro* (Wang et al., 2024c); 4200 examples, including 300 randomly sampled questions per domain (e.g., physics, law, computer science) extends the original MMLU benchmark (Hendrycks et al., 2021a) by adding reasoning-focused questions and expanding the choice set from four to ten. (3) *HotpotQA* (Yang et al., 2018); 500 samples from (Kim et al., 2024) a multi-hop question-answering benchmark requiring reasoning across multiple Wikipedia paragraphs with annotated supporting facts. (4) *DROP* (Dua et al., 2019); 500 randomly sampled questions from this reading comprehension benchmark, demanding discrete numerical reasoning (e.g., addition, counting, sorting) over paragraphs. (5) GPQA-diamond (Rein et al., 2024) (*GPQA*); 198 graduate-level questions assessing complex reasoning in biology, physics, and chemistry. (6,7) MuSR (Sprague et al., 2024); 256 examples in Object Placement (*MuSR-op*) and 250 examples in Team Allocation (*MuSR-ta*) tasks assessing spatial and relational reasoning.

Notably, prior work (Sprague et al., 2025) has shown that few-shot Chain-of-Thought (CoT) prompting yields significant gains over zero-shot CoT in MuSR, highlighting the role of in-context examples in complex reasoning. As few-shot examples are necessary for some baselines and ReFeri, we collect them following the previous works. MATH500: 5 examples from (Yang et al., 2024) (GPTs), 4 examples from (Lewkowycz et al., 2022) (LLaMA).[3] MMLU-Pro: 5 examples from (Wang et al., 2024c). HotpotQA: 6 examples from (Yao et al., 2023). DROP: 3 examples following (Zhou et al., 2022). GPQA-Diamond: 5 examples from (Rein et al., 2024). MuSR: 3 examples from (Sprague et al., 2025). Complete prompt templates are available in Appendix A.1.

**Baselines.** We compare ReFeri against five widely-used prompt-based methods that require no additional training, with some reflecting different uses of few-shot examples: (1) *Zero-shot CoT* appends a trigger phrase ("Let's think step by step.") to each query without providing exemplars, instead relying on LLM's intrinsic reasoning capabilities. (2) *Few-shot CoT* prepends a fixed set of few examples, enabling LLM to generalize from few in-context demonstrations. (3) *LEAP* (Zhang et al., 2024) improves few-shot prompting by intentionally inducing mistakes on few examples. Then extracting generalizable task-specific principles through self-reflection without human annotations, and prompting the model to apply these principles to unseen questions. Specific prompts for each baseline are in Appendix A.2. (4) *USC* asks LLM to select the best answer from multiple CoT outputs, by following (Chen et al., 2023). (5) *CoT-WP* (Wang & Zhou, 2024) scores each candidate response using token-level probabilities from LLM conditioned on the same few-shot examples. Specifically, the score is a confidence gap between top-1 and top-2 tokens at answer positions.

**Implementation details.** For the experiments, we use (1) `gpt-4o-2024-08-06` (*GPT-4o*) (OpenAI, 2024a), (2) `gpt-4o-mini-2024-07-18` (*GPT-4o-mini*) (OpenAI, 2024b), and (3) `LLaMA-3.1-8B-Instruct` (*LLaMA-3.1-8B*) (Dubey et al., 2024) as target LLMs, *i.e.,* response generation models. We generate $K = 5$ responses per query using temperature of 1.0 to encourage diverse candidates. For Zero-shot CoT, Few-shot CoT and LEAP, we report the average accuracy across five responses without applying any selection mechanism, which can be viewed as randomly selecting the response. For USC, CoT-WP and ReFeri, we use the same candidates generated from Few-shot CoT and employ LLaMA-3.1-8B-Instruct as the estimation (or LLM-judge) model, except in the experiments Figure 4 and Table 17. For all results, estimation model's temperature is fixed at

---

[3](1) Using the same prompt as GPT results in significantly lower accuracy, and (2) LLaMA-based models provide their own optimized prompt templates (see `meta-llama/Llama-3.2-3B-Instruct-evals`).

Table 1: **Main results.** Overall performance on seven reasoning benchmarks comparing the proposed **ReFeri** with different baselines not require additional training, under three different state-of-the-art LLMs. The best and second-best scores are highlighted in **bold** and underline, respectively.

| Models | Methods | MuSR-ta (Acc.) | MuSR-op (Acc.) | GPQA (Acc.) | MATH500 (Acc.) | DROP (EM / F1) | HotpotQA (EM / F1) | MMLU-PRO (Acc.) | Avg. |
|---|---|---|---|---|---|---|---|---|---|
| LLaMA-3.1-8B | Zero-shot CoT | 43.0 | 50.6 | 21.6 | 44.2 | 60.4 / 66.4 | 15.2 / 21.2 | 39.8 | 39.3 |
| | Few-shot CoT | 64.8 | 53.3 | 24.0 | 42.9 | 61.4 / 67.3 | 19.0 / 25.1 | 38.7 | 43.4 |
| | LEAP | 69.2 | 51.6 | 27.8 | 42.3 | 58.2 / 64.1 | 19.9 / 26.8 | 37.3 | 43.8 |
| | USC | 67.2 | 52.3 | 28.8 | 49.6 | 69.6 / 75.8 | 24.4 / 32.5 | 45.6 | 48.2 |
| | CoT-WP | 72.4 | 54.7 | 29.3 | 47.8 | **71.6** / 75.8 | **25.8** / **33.4** | **46.0** | 49.7 |
| | ReFeri | **79.6** | **57.8** | **35.4** | **51.2** | 69.4 / 75.7 | 25.0 / 33.2 | 45.1 | **51.9** |
| GPT-4o-mini | Zero-shot CoT | 56.2 | 58.1 | 43.0 | 76.4 | 77.6 / **85.6** | 31.5 / 41.4 | 63.0 | 58.0 |
| | Few-shot CoT | 77.0 | 59.4 | 41.3 | 75.2 | 76.8 / 83.1 | 34.0 / 45.1 | 63.0 | 61.0 |
| | LEAP | 74.4 | 59.8 | 43.9 | 74.5 | 75.8 / 83.0 | 34.0 / 45.1 | 63.2 | 60.8 |
| | USC | 74.4 | 60.9 | **46.0** | 77.8 | 76.8 / 83.8 | 35.0 / 47.2 | 63.7 | 62.1 |
| | CoT-WP | 78.8 | 56.3 | 42.4 | 77.8 | 76.4 / 82.5 | 35.8 / 46.7 | 64.6 | 61.7 |
| | ReFeri | **82.8** | **61.3** | 41.9 | 77.8 | **79.2** / 84.9 | **36.2** / **48.0** | **64.9** | **63.4** |
| GPT-4o | Zero-shot CoT | 66.6 | 61.7 | 48.8 | 77.5 | 75.1 / 85.3 | 37.6 / 49.9 | 73.9 | 63.0 |
| | Few-shot CoT | 87.0 | 69.7 | 47.8 | 75.6 | 80.6 / 89.2 | 44.6 / 58.4 | 73.7 | 68.4 |
| | LEAP | 87.2 | 66.8 | 45.5 | 75.6 | 81.5 / 89.8 | 45.1 / 58.4 | 74.0 | 68.0 |
| | USC | 85.2 | 71.1 | 47.0 | 77.4 | 82.2 / 90.2 | 45.6 / 59.7 | 74.5 | 69.0 |
| | CoT-WP | 88.0 | 68.8 | 47.5 | **78.4** | 83.4 / **91.4** | **47.2** / 60.2 | 74.1 | 69.6 |
| | ReFeri | **90.4** | **71.9** | **51.5** | 77.8 | **83.6** / 91.1 | 47.0 / **60.7** | **75.4** | **71.1** |

1.0. In USC (*i.e.*, LLM-as-Judge setting), the decoding temperature is fixed at 0 for determinism. For computing similarity in backward consistency score (Eq. 9), we employ the lightweight embedding model `all-mpnet-base-v2` with 110M parameters. More details are in Appendix A.3.

## 3.2 MAIN RESULTS

Table 1 summarizes the experimental results across seven different reasoning benchmarks and three different LLMs. For instance, across all LLMs and benchmarks, ReFeri improves average accuracy by 4.5% over Few-shot CoT, which corresponds to apply random selection instead. Compared to the second-best method, CoT-WP, ReFeri achieves an average improvement of 1.8% across all benchmarks. Notably, CoT-WP relies solely on the forward likelihood of each candidate, while ReFeri combines both forward and backward signals via a Bayes-derived scoring function. This bidirectional formulation allows ReFeri to capture not just the confidence of an answer, but also its consistency with few-shot examples upon the LLM's intrinsic knowledge about the task; consequently, it enables a better selection across various tasks. We note that performance of prompt-based selection, USC, largely varies depending on the task and used LLMs, which reveals the limitation of solely relying on LLM's intrinsic knowledge. In addition, as mentioned in Section 3.1, MuSR is a benchmark where few-shot examples play a critical role (Sprague et al., 2025) and our results also support this with 21.0% average improvement by Few-shot CoT over Zero-shot CoT. Here, we find that ReFeri further enlarges the improvement with the largest gain, outperforming the second-best method by 4.5%. This result shows that ReFeri is particularly effective in new domains where LLM has little prior knowledge and need to heavily rely on a few examples without additional training or reward models.

Next, to assess whether ReFeri scales effectively with the number of candidate outputs similar to the conventional reward-based best-of-$K$ selection, we evaluate performance as the candidate pool grows. Specifically, we test $K = \{1, 5, 10, 15, 20\}$ candidates on three representative tasks—*MATH500*, *GPQA*, and *MuSR-ta* by using `GPT-4o-mini` as the generation model under Few-shot CoT. Across the three tasks, ReFeri yields consistent improvements as $K$ increases. On MATH500, while the accuracy of random selection decreases as the number of generated samples increases, ReFeri consistently selects higher-quality responses, improving from 75.8% at $K = 1$ to 79.4% at $K = 20$. On GPQA, where ReFeri raises performance from 41.4% to 45.5% as the candidate pool grows. Consistently, the largest gain is observed on MuSR-ta, which saw a sharp jump in accuracy from 75.6% to 86.0%, an improvement of 10.4%. In contrast, CoT-WP and USC exhibit unstable accuracy under the test-time scaling. Their performance even degrades as the number of candidates increases, suggesting that these methods do not capture what is truly plausible among the candidates. Notably, USC demonstrates strong performance on GPQA when K=5, but its accuracy declines as K increases, highlighting sensitivity to the candidate set size. In addition, we observe an inherent ordering bias in USC: selections come from the first two responses regardless of correctness (see Appendix B.1), indicating a limitation of prompt-based approach. Overall, these results confirm that ReFeri scales well with more candidates, demonstrating effectiveness and reliability in practical test-time scaling.

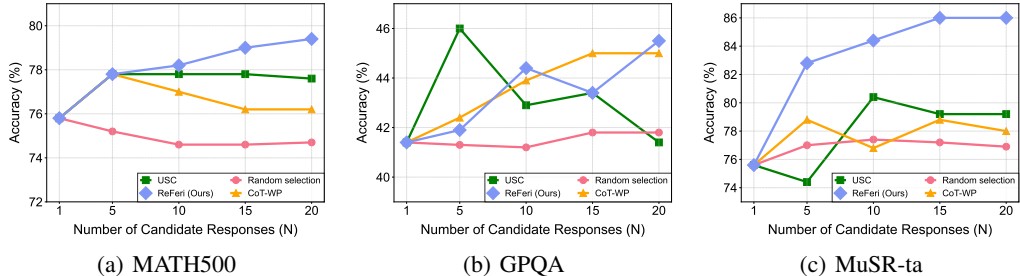

(a) MATH500                    (b) GPQA                    (c) MuSR-ta

Figure 3: **Test-time scaling with ReFeri.** Accuracy of ReFeri versus other training-free selection methods (Random selection, CoT-WP and USC) on MATH500, GPQA, and MuSR-ta. GPT-4o-mini generate different numbers of candidate responses ($K = 1, 5, 10, 15, 20$) using Few-shot CoT.

Table 2: **Ablation study.** Evaluation of scoring variants averaged over three generation LLMs, comparing the contribution of metric term (forward and backward) on MATH500 and GPQA.

Table 3: **Different few-shot examples.** Accuracy across three different choices of few-shot examples on MATH500 (top) and GPQA (bottotm) using GPT-4o-mini to generate responses.

|  | Forw. | Back. | MATH500 | GPQA |
|---|---|---|---|---|
|  | ✗ | ✗ | 64.6 | 37.7 |
|  | ✓ | ✗ | 68.9 | 42.2 |
|  | ✗ | ✓ | 64.6 | 34.7 |
| ReFeri (Full) | ✓ | ✓ | 69.0 | 42.7 |
| ReFeri | ✓ | ✓ | 68.9 | 42.9 |

| Methods | 1st | 2nd | 3rd | Avg. |
|---|---|---|---|---|
| Few-shot CoT | 75.2 | 74.5 | 75.0 | 74.9 |
| ReFeri (Ours) | 77.8 | 79.0 | 77.8 | 78.2 |
| Few-shot CoT | 41.3 | 41.5 | 38.9 | 40.6 |
| ReFeri (Ours) | 41.9 | 43.4 | 41.9 | 42.4 |

## 3.3 ADDITIONAL ANALYSES

In this section, we present additional analyses of ReFeri. We primarily perform experiments with GPT-4o-mini on MATH500 and GPQA datasets. More results are presented in Appendix B.

**Ablation study.** To better understand which components contribute to the effectiveness of ReFeri, we conduct an ablation study on each part of the proposed scoring function (Eq. 11), which is grounded in Bayes' rule (Eq. 5). In Table 2, we report performance on MATH500 and GPQA, averaged across the three LLMs used in Table 1. First, one can observe that using the combined score yields better results compared to solely using the forward confidence score (Eq. 6) or the backward consistency score (Eq. 10). This complementary effect comes from their different natures; while the forward score focuses on model-generated response which may contain noise, backward score utilizes given few-shot examples, which are well-curated inputs and ground-truth labels, and thus relatively clean. Next, it is also observed that using the cost-efficient variant of backward term *(ReFeri)* does not compromise the performance compared to *ReFeri (Full)* which uses original backward score (Eq. 8), when combined with forward term (more results are in Table 18). This result mitigates concerns regarding the additional computations incurred by original backward term.

**Robustness to few-shot examples.** ReFeri highly relies on few-shot examples for scoring of both forward and backward scores (Section 2.2). This raises the question of how sensitive the method is to the choice of few-shot examples. To answer this, we conduct a sensitivity study on MATH500 and GPQA using GPT-4o-mini, where we use three different few-shot examples with one original and two newly sampled. As shown in Table 3, both Few-shot CoT and ReFeri show variation across these different sets. Nevertheless, ReFeri consistently outperforms Few-shot CoT which corresponds to random selection, and the average gap remains 2.6%. These results indicate that ReFeri remains robust to exemplar choice and is consistently effective, rather than overfitted to specific demonstrations.

Moreover, in practical applications, the clarity of few-shot example might be not always guaranteed. To verify the effectiveness of ReFeri under this scenario, we first synthesize low-quality few-shot examples by converting the original examples via prompting GPT-4o-mini to degrade the quality of reasoning in data. The degradation of quality is indeed confirmed through LLM-as-judge framework (results and judgments are in Appendix B.6 ) As shown in Table 13, ReFeri maintains consistent improvements even under degraded exemplars, indicating that recycled few-shot examples as verification remains effective without well curated examples; for instance, on MATH500 with LLaMA-3.1-8B, accuracy improves from 39.5% to 47.6% (+8.1), demonstrating ReFeri's robustness.

Additionally, to investigate the impact of various prompt choices, we conduct new experiments with two alternative prompting techniques, following prior work planning and role-playing (Wang et al., 2023a; Kong et al., 2024). Specifically, we assess the robustness of ReFeri by varying prompts during both the generation stage and the verification stage by adapting different prompting styles (orig, plan, and role). For plan and role prompting at the generation, we newly sample five responses similar to Table 1. Table 4 shows the performance on MATH500 and GPQA under

Table 4: **Ablation on generation/evaluation prompts.** Evaluation on MATH500 and GPQA with generation/evaluation prompt variants (orig/plan/role).

| Gen | Eval | MATH500 | | GPQA | |
|---|---|---|---|---|---|
| | | Few-shot | ReFeri | Few-shot | ReFeri |
| **Orig** | Orig | 75.2 | **77.8** | 41.3 | **41.9** |
| | Plan | 75.2 | **78.0** | 41.3 | **42.4** |
| | Role | 75.2 | **77.8** | 41.3 | **41.9** |
| **Plan** | Plan | 74.6 | **78.2** | 42.6 | **47.5** |
| | Orig | 74.6 | **78.4** | 42.6 | **47.5** |
| **Role** | Role | 74.5 | **78.2** | 43.5 | **47.5** |
| | Orig | 74.5 | **78.2** | 43.5 | **47.0** |

different configurations. A key observation is that verification performance remains highly consistent across different verification prompt styles for a given generation prompt style. For instance, on GPQA, performance for "plan → orig" and "plan → plan" conditions is identical (47.5 vs. 47.5), with similar consistency observed for the "role" condition (47.5 vs. 47.0). This indicates that ReFeri is inherently robust to variations in prompt formatting during the evaluation stage.

However, we also observe that the initial quality of the generated candidate set varies depending on the prompt style. For relatively simple tasks like MATH500, the quality of generated responses is similar across prompts. Conversely, on the more challenging GPQA, prompts offering structured guidance (*e.g.*, plans or roles) tend to generate higher-quality seeds, reflected in slightly higher accuracy. Consequently, ReFeri performs better when the initial candidates are of higher quality.

**Estimation models and computational cost.** To examine whether ReFeri is robust to the choice of estimation model $p_\theta$, we evaluate its performance using three LLMs with diverse scales and backbones: `LLaMA-3.2-1B-Inst`, `Qwen-2.5-7B-Inst`, and `LLaMA-3.1-70B-Inst`. The generation model is fixed (either GPT-4o-mini, GPT-4o, or LLaMA-3.1-8B), and we apply each estimation models to two tasks on MATH500 and GPQA. The average accuracy of three generation LLMs is presented in Figure 4 (Full results are in Appendix B.8). Here, ReFeri consistently improves Few-shot CoT across all settings, with an average gain of 4.9% on MATH500 and 5.1% on GPQA. Notably, the smallest model (LLaMA-3.2-1B) performs competitively, and even achieves competitive

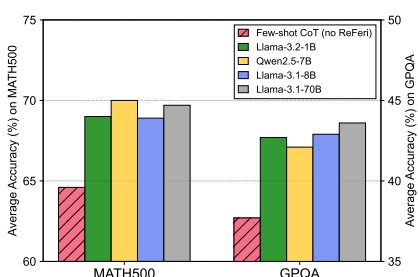

Figure 4: **Estimation model.** Each bar shows the average accuracy of three generation LLMs on MATH500 and GPQA.

performance on MATH500. We attribute this to the relative simplicity of MATH benchmark, as recent small LLMs often exhibit reasonable performance; hence, they can make reliable likelihood estimates for selection. In contrast, GPQA requires more complex reasoning; therefore, using the large estimation model could be more beneficial. Indeed, LLaMA-3.1-70B achieves the best performance on this case. Despite these task-specific differences, the overall improvements are consistent across all estimation models. This suggests that the effectiveness of ReFeri primarily stems from its validation strategy with few-shot examples, rather than the specific choice of estimation model.

In addition, this consistent effectiveness offers better cost-accuracy trade-off. To show this, we conduct experiments using a small verifier for ReFeri (LLaMA-3.2-1B) and compared it against baselines that rely on a larger model (USC and CoT-WP with LLaMA-3.1-8B). Accuracy and latency per query (seconds per instance on a single GPU with identical configuration) are reported in Table 17. Here, ReFeri with a 1B estimator outperforms the strong 8B CoT-WP baseline while substantially reducing latency. For instance, on MATH500, ReFeri (1B) is approximately 60–65% faster than the CoT-WP (8B) baseline (*e.g.*, 3.0s vs. 8.3s on MATH500). Furthermore, ReFeri exhibits robust performance regardless of estimation model size, whereas baselines often suffer significant degradation when scaled down. This demonstrates that combining ReFeri with a small-scale estimator provides a highly advantageous, delivering robust validation at remarkably low computational cost.

## 4    RELATED WORKS

**Few-shot in-context learning of LLM.** Few-shot in-context learning (ICL) revealed that LLMs can generalize to unseen tasks with just a handful of input-output demonstrations (Brown et al., 2020). To handle complex reasoning problems, chain-of-thought (CoT) prompting was proposed to append intermediate steps to the few-shot examples, leading to substantial gains in tasks such as arithmetic, commonsense reasoning, and symbolic manipulation (Wei et al., 2022; Fu et al., 2023; Jin et al., 2024). To further enhance ICL, various strategies have been developed to retrieve better examples using semantic similarity or entropy-based selection (Wu et al., 2023; Peng et al., 2024). However, some studies have shown that few-shot ICL does not always guarantee improvements. For instance, label shuffling or format changes can often leave performance unaffected (Min et al., 2022), and the performance gap between zero-shot and few-shot CoT is narrowing in several benchmarks as instruction tuning becomes more effective (Sprague et al., 2025). In particular, recent LLMs such as DeepSeek-R1, which are trained with reinforcement learning-based reasoning steps, sometimes even show performance degradation when few-shot CoT examples are added (Guo et al., 2025). Nonetheless, carefully selected demonstrations are still effective (Huang et al., 2024). For example, (Ge et al., 2025) show that few-shot examples can reduce overconfidence in multi-step reasoning, and (Yan et al., 2025) show that they help mitigate hallucinations and memory-based mistakes in complex tasks. These observations motivate us to go beyond using few-shot examples solely for generation, and recycling them to evaluate multiple LLM responses and to select the most promising one.

**Selection of diverse LLM outputs.** Due to the probabilistic nature of LLM decoding, LLM can provide diverse outputs for a single input, each reflecting different reasoning paths (Kadavath et al., 2022; Wang & Zhou, 2024; Qiu & Miikkulainen, 2024; Kang et al., 2025). To handle this variability, self-consistency (Wang et al., 2023b) samples $K$ independent reasoning paths and selects the majority answer to improve accuracy. However, it assumes that the model produces a single, well-formatted answer, and this assumption is often violated in open-ended tasks such as summarization or free-form dialogue (Stiennon et al., 2020; Salemi et al., 2024). Alternatively, recent Best-of-N approaches aim to directly select the best output among candidates, often using external verification models. For instance, Outcome Reward Models (ORMs) grade final outputs (Cobbe et al., 2021; Uesato et al., 2022), while Process Reward Models (PRMs) assess intermediate reasoning steps to provide finer supervision (Lightman et al., 2024; Wang et al., 2024b). Despite their successes, these models require large-scale, task-specific annotations or domain-specific checkers, limiting their scalability to new domains or unseen tasks. To eliminate the need for external verification models, prompting-based methods such as LLM-as-Judge ask LLM to evaluate its own outputs (Chen et al., 2023; Zheng et al., 2023). However, their effectiveness heavily depends on the model's prior knowledge in the target domain. When this knowledge is lacking, these methods require additional fine-tuning with curated evaluation datasets for sufficient performance, which reintroduces the need for supervision (Yuan et al., 2024; Mahan et al., 2024; Zhang et al., 2025). In contrast, ReFeri is training-free and task-agnostic, offering a more scalable and generalizable approach by recycling a few-shot examples for verification.

## 5    CONCLUSION

We propose ReFeri, a training-free framework to find promising LLM output by reusing few-shot data not only for generation but also for verification. In experiments, ReFeri performs consistently effective in various LLMs and tasks, demonstrating robustness across few-shot data and prompt variations. It suggests that ReFeri is a practical way to find the reliable LLM output with minimal human involvement, opening future directions to reconsider the broader utility of few-shot examples.

**Limitation and future works.** Since the selection by ReFeri is determined by likelihoods produced by an estimation model, it does not explain why a response is incorrect, unlike PRMs, which offer step-level feedback, or LLM-as-judge, which can easily generate explanations by prompting. However, we believe that ReFeri can potentially provide a certain level of interpretability; for example, we visualize the token-level uncertainty of candidate responses and observe that it reveals potentially untrustworthy tokens (see Appendix D). This kind of token-level consideration not only provides the interpretability but also can improve the effectiveness of ReFeri, suggesting a future direction.

## ETHICS STATEMENT

ReFeri provides a training-free method for selecting promising outputs from LLMs. This makes it particularly valuable in scenarios where labeled data is scarce or where model fine-tuning is impractical such as limited access to data, or applications in emerging domains where predefined labels are unavailable. In addition, ReFeri reduces the barrier to deploying LLMs in real-world settings without additional supervision. This may contribute to broader and more efficient adoption of LLMs in resource-constrained environments. All datasets used are public and widely adopted.

## REPRODUCIBILITY STATEMENT

For reproducibility, we provide detailed prompts, datasets, and experimental setups in Appendix A. In Section 3 and Appendix B, we report extensive experiments that demonstrate the robustness of our approach. In addition, we will release our code to ensure transparency and facilitate further research.

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

# A  MORE DETAILS OF EXPERIMENTAL SETUPS

This section covers more details about the experiments from Section 3.

## A.1  DATASETS

This subsection provides more information about the dataset and the few-shot examples we used.

- **MATH500.** The MATH benchmark (Hendrycks et al., 2021b) consists of 12,500 LaTeX-formatted competition-level math problems, with topics ranging from algebra and geometry to number theory. Each problem includes a step-by-step solution and expects the model to generate a boxed final answer (e.g., an integer or simplified expression). We use MATH500, a 500-question subset introduced in (Lightman et al., 2024), uniformly sampled from the test split to preserve subject and difficulty distribution. For few-shot examples, we follow (Yang et al., 2024)[4] for GPT-based models and (Lewkowycz et al., 2022)[5] for LLaMA-based models. The reason for this choice is based on our empirical observation: Simply adding "Please think step by step and put your final answer within \boxed{}." as done in GPT-style few-shot prompts led to a significant drop in accuracy. Namely, LLaMA-based models require prompt formats that are aligned with their own instructions and are sensitive to deviations from the learned template. This benchmark evaluates symbolic reasoning ability in mathematical domains.

- **MMLU-Pro.** MMLU-Pro (Wang et al., 2024c) is an extension of the original MMLU benchmark (Hendrycks et al., 2021a), which evaluates broad knowledge and reasoning over 57 subjects using 14k 4-way multiple-choice questions. MMLU-Pro introduces 12k curated 10-way multiple-choice questions across 14 professional domains, increasing task difficulty and emphasizing complex, multi-step reasoning. Instead of using the full test set, we subsample 300 questions per subject (totaling 4,200) using random seed 42 and we will share the used indices at the code. For few-shot examples, we follow the format used in (Wang et al., 2024c). This benchmark is used to assess domain-specific and robust reasoning performance.

- **GPQA.** GPQA (Rein et al., 2024) is a graduate-level QA benchmark consisting of 448 expert-authored multiple-choice questions in domains such as physics, chemistry, and biology. Designed to be "Google-proof," it focuses on evaluating complex scientific reasoning that cannot be answered through simple retrieval. We evaluate on GPQA-Diamond, a curated subset of 198 especially difficult questions selected by the authors. Few-shot examples are taken directly from the official release (Rein et al., 2024). This task measures deep scientific understanding.

- **DROP.** The DROP benchmark (Dua et al., 2019) contains 96k question-answer pairs requiring discrete reasoning over Wikipedia passages (e.g., numerical operations, counting, or date comparison). Answers may include spans, numbers, or dates. We evaluate on a 500-sample subset randomly selected from the dev set, and we will share the selected indices at the code. We use 3-shot examples from (Zhou et al., 2022) and report both EM and F1 metrics following the official implementation. This benchmark evaluates models' symbolic reasoning grounded in natural language passages.

- **HotpotQA.** HotpotQA (Yang et al., 2018) consists of 113k multi-hop QA pairs requiring reasoning over multiple Wikipedia documents. The model must retrieve at least two relevant passages and combine facts to answer each question. We follow the (Kim et al., 2024), which uses 500 samples from the dev set. Few-shot examples are taken from (Yao et al., 2023). This task tests compositional reasoning and the ability to aggregate distributed information across documents.

- **MuSR.** MuSR (Sprague et al., 2024) is a benchmark for multi-step reasoning over long-form narratives (800–1000 words), constructed via neuro-symbolic generation to embed logical dependencies into natural language. It includes structured tasks such as TeamAllocation (constraint-based planning) and ObjectPlacement (spatial consistency reasoning). We evaluate on the 256 TeamAllocation and 250 ObjectPlacement examples from the official release (Sprague et al., 2024), using 3-shot prompts tailored to each task (Sprague et al., 2025). MuSR requires understanding of narrative flow, contextual logic, and physical feasibility. As demonstrated in (Sprague et al., 2025), ICL plays a critical role in model performance on MuSR, and demonstrates the effectiveness of ReFeri.

---

[4] https://github.com/QwenLM/Qwen2.5-Math
[5] https://huggingface.co/datasets/meta-llama/Llama-3.2-3B-Instruct-evals

## A.2 BASELINES

Here, we provide the template used for our baseline, using MATH500 as a representative task among multiple benchmarks. (see list 1–7).

## A.3 IMPLEMENTATION

This section provides the detailed information needed to implement the main experiment.

**Resource details.** To avoid out-of-memory, we used two NVIDIA H100 GPUs for evaluation with the LLaMA-3.1-70B-Instruct model. All other experiments were performed on a single A6000 GPU.

**Response generation.** We use lm-eval-harness[6] to generate responses from LLaMA-based models, with temperature set to 1.0 and 5 responses per input. The prompt was written in chat template format using vllm.[7] For GPT-family models, we use the official OpenAI API to generate completions under the same sampling configuration. The remaining settings follow the GPT API default settings. During evaluation, we report the average score across the 5 generations. All evaluations are conducted using our custom evaluation scripts to ensure consistent scoring and formatting across models.

**Algorithm of ReFeri.** In algorithm 1, we present the formal algorithm for ReFeri. We generate multiple candidate responses $\{r_1, ..., r_K\}$ for each test query using Few-shot CoT, as it exhibit the better quality on the average (see Table 1).

---

**Algorithm 1** ReFeri algorithm

---

**Input:** estimation model $P$, embedding model $E$, test-query $\widetilde{q}$, $K$ candidate responses $\{r_1, \ldots, r_K\}$, $N$ few-shot examples $\mathbf{X} = \{\mathbf{x}_i\}_{i=1}^N$, replaced prompt $\widetilde{\mathbf{X}}_i$

---

$i^\dagger = \arg\max_{i=1,\ldots,N} \cos\left(E(\widetilde{q}), E(q_i)\right)$
**for** $k = 1$ **to** $K$ **do**
    $S_{\texttt{Forw}} \leftarrow$ Compute forward score with $r_k$ as label, using $P$ and context $(\widetilde{q}, \mathbf{X})$ (Eq. 6)
    Construct $\widetilde{\mathbf{X}}_{i^\dagger} \leftarrow$ using a leave-one-out strategy (Eq. 7)
    $S_{\texttt{Back}} \leftarrow$ backward score with $a_i$ as label, using $P$ and $\widetilde{\mathbf{X}}_{i^\dagger}$ (Eq. 10)
    $S_{\texttt{Fin}} \leftarrow S_{\texttt{Forw}} - S_{\texttt{Back}}$ (Eq. 11)
    $S_k \leftarrow S_{\texttt{Fin}}$
**end for**
$r_{k^*} \leftarrow \arg\max_k S_k$ (Eq. 3)
**return** $r_{k^*}$

---

## B MORE QUANTITATIVE RESULTS

### B.1 ADDITIONAL COMPARISON WITH FEW-SHOT PROMPTING-BASED SELECTION METHODS

Among the multiple answer selection methods, the simplest and most accessible approach (*e.g.*, learning overhead, domain specificity, etc.) is arguably LLM-as-Judge Chen et al. (2023); Zheng et al. (2023). It uses the LLM itself to score and select answers via in-context learning without any additional training or external verifiers. In particular, the addition of few-shot examples to LLM-as-Judge might be most closely aligned with the core motivation of ReFeri, which is to use demonstrations not only for generation but also for validation. Therefore, in this section, we compare ReFeri and (1) the original *USC* (Chen et al., 2023), (2) *USC with few-shot* (our adaptation), and (3) *LLM-as-Judge with few-shot* created with our optimized prompt (see list 8 and 9).

As shown in Table 5, ReFeri consistently achieves the best or second-best accuracy across all LLMs and benchmarks. Interestingly, we observe that adding few-shot demonstrations to USC often degrades performance (*e.g.*, on GPQA and DROP with GPT-4o-mini and LLaMA-3.1-8B), which is likely due to the sensitivity of LLMs to prompt format and positional bias of the responses.

---

[6] https://github.com/EleutherAI/lm-evaluation-harness
[7] https://huggingface.co/datasets/meta-LLaMA/LLaMA-3.1-8B-Instruct-evals

Table 5: **Comparison with prompting-based selection.** Overall performance on seven reasoning benchmarks comparing the proposed ReFeri with different prompting-based baselines not require additional training, under three different state-of-the-art LLMs. For reference, we additionally include the Oracle upper bound.

| Models | Methods | MuSR-ta | MuSR-op | GPQA | MATH500 | DROP | HotpotQA | MMLU-PRO | Avg. |
|---|---|---|---|---|---|---|---|---|---|
| LLaMA-3.1-8B-It | USC | 67.2 | 52.3 | 28.8 | 49.6 | **69.6** / 75.8 | 24.4 / 32.5 | **45.6** | 48.2 |
| | USC-w/Fewshot | 70.0 | 53.9 | 28.3 | 47.8 | 69.0 / 75.3 | **25.2** / 32.3 | 45.1 | 48.5 |
| | LLM-as-Judge | 65.2 | 55.1 | 21.2 | 46.0 | 67.7 / 74.0 | 23.4 / 31.2 | 44.1 | 46.1 |
| | ReFeri | **79.6** | **57.8** | **35.4** | **51.2** | 69.4 / 75.7 | 25.0 / **33.2** | 45.1 | **51.9** |
| | Oracle | 97.6 | 88.3 | 59.6 | 66.6 | 83.4 / 88.8 | 33.8 / 45.0 | 70.8 | 71.4 |
| GPT-4o-mini | USC | 74.4 | 60.9 | **46.0** | 77.8 | 76.8 / 83.8 | 35.0 / 47.2 | 63.7 | 62.1 |
| | USC-w/Fewshot | 76.4 | **63.3** | 39.9 | **78.2** | 77.2 / 84.0 | 34.8 / 46.6 | 63.2 | 61.9 |
| | LLM-as-Judge | 75.6 | 60.6 | 34.3 | 77.0 | 77.4 / 84.4 | 35.0 / 46.7 | 63.3 | 60.5 |
| | ReFeri | **82.8** | 61.3 | 41.9 | 77.8 | **79.2** / **84.9** | **36.2** / **48.0** | **64.9** | **63.4** |
| | Oracle | 97.2 | 78.1 | 70.7 | 85.8 | 86.6 / 91.6 | 43.4 / 56.4 | 76.8 | 76.9 |
| GPT-4o | USC | 85.2 | 71.1 | 47.0 | 77.4 | 82.2 / 90.2 | 45.6 / 59.7 | 74.5 | 69.0 |
| | USC-w/Fewshot | 88.8 | 69.1 | 46.0 | 77.4 | 82.0 / 89.9 | 45.4 / 60.1 | 74.1 | 69.0 |
| | LLM-as-Judge | 86.0 | 68.0 | 46.5 | **77.8** | 82.8 / 91.0 | 45.6 / 59.8 | 73.3 | 68.6 |
| | ReFeri | **90.4** | **71.9** | **51.5** | **77.8** | **83.6** / **91.1** | **47.0** / **60.7** | **75.4** | **71.1** |
| | Oracle | 96.4 | 87.9 | 72.2 | 86.6 | 89.2 / 94.7 | 55.4 / 69.2 | 84.1 | 81.7 |

Table 6: **Response selection distribution per task (GPT-4o-mini).**

| Task | Method | #1 | #2 | #3 | #4 | #5 | Fail (-1) |
|---|---|---|---|---|---|---|---|
| MATH500 | USC | 90.2 | 3.8 | 1.0 | 2.8 | 2.2 | 0.0 |
| | USC-w/ Fewshot | 81.0 | 12.6 | 0.6 | 3.2 | 2.6 | 0.0 |
| | LLM-as-Judge | 51.4 | 2.2 | 8.0 | 12.2 | 25.4 | 0.8 |
| MMLU-Pro | USC | 34.2 | 19.1 | 7.2 | 24.0 | 15.5 | 0.0 |
| | USC-w/ Fewshot | 18.8 | 31.0 | 7.8 | 22.7 | 19.7 | 0.0 |
| | LLM-as-Judge | 22.1 | 8.0 | 7.8 | 13.5 | 48.3 | 1.2 |
| GPQA | USC | 21.7 | 15.2 | 13.1 | 23.7 | 26.3 | 0.0 |
| | USC-w/ Fewshot | 19.7 | 17.7 | 10.6 | 23.7 | 28.3 | 0.0 |
| | LLM-as-Judge | 30.8 | 9.1 | 10.1 | 7.6 | 42.4 | 0.0 |
| DROP | USC | 73.8 | 21.8 | 1.8 | 1.2 | 1.4 | 0.0 |
| | USC-w/ Fewshot | 78.2 | 16.6 | 3.0 | 1.0 | 1.2 | 0.0 |
| | LLM-as-Judge | 68.8 | 9.6 | 4.6 | 4.4 | 12.4 | 0.2 |
| HotpotQA | USC | 77.0 | 13.4 | 2.8 | 5.4 | 1.4 | 0.0 |
| | USC-w/ Fewshot | 68.6 | 20.6 | 2.8 | 6.0 | 2.0 | 0.0 |
| | LLM-as-Judge | 65.0 | 15.0 | 4.8 | 6.0 | 9.2 | 0.0 |
| MuSR-op | USC | 51.6 | 18.0 | 11.7 | 7.4 | 11.3 | 0.0 |
| | USC-w/ Fewshot | 36.7 | 40.2 | 10.2 | 6.2 | 6.6 | 0.0 |
| | LLM-as-Judge | 26.9 | 21.5 | 15.2 | 20.7 | 15.2 | 0.4 |
| MuSR-ta | USC | 34.0 | 3.2 | 0.8 | 9.6 | 9.6 | 42.8 |
| | USC-w/ Fewshot | 46.4 | 35.2 | 2.4 | 3.6 | 12.4 | 0.0 |
| | LLM-as-Judge | 27.6 | 2.8 | 0.4 | 1.2 | 50.4 | 17.6 |

Notably, we observe that both prompt-based selection methods, USC and LLM-as-Judge, are highly sensitive to the order of candidate responses. In our experiments, USC frequently selections were made from the first two responses regardless of correctness; on multiple choice question tasks this pattern is less extreme but skew toward early positions is still visible. Moreover, since USC requires explicit answer extraction, tasks such as MuSR-ta revealed many failure cases (*e.g.*, over 40% failures in Table 6), further highlighting its fragility. This highlights a critical weakness in prompt-based selection: the output is often determined more by position than content. Based on these observations, we conducted additional experiments where we randomly rearranged the order of candidate responses. Indeed, we observed this issue in Table 7; on GPQA, for example, USC's accuracy varied notably across different permutations (*e.g.*, 46.0 → 41.9), demonstrating its sensitivity to presentation order. In contrast, our approach mitigates such ordering artifacts by decoupling few-shot demonstrations

Table 7: **Evaluation of USC ordering with GPT-4o-mini.** Two random permutations (*perm-A*, *perm-B*) of the candidate order versus the original order.

| Methods | MuSR-ta | MuSR-op | GPQA | MATH500 | DROP | HotpotQA | MMLU-PRO | Avg. |
|---|---|---|---|---|---|---|---|---|
| USC (perm-A) | 75.6 | 59.8 | 41.9 | 78.0 | 76.6 / 83.5 | 35.6 / 47.2 | 63.2 | 61.5 |
| USC (perm-B) | 77.2 | 56.6 | 45.0 | 77.2 | 76.6 / 83.4 | 35.2 / 46.9 | 63.6 | 61.6 |
| USC (default) | 74.4 | 60.9 | 46.0 | 77.8 | 76.8 / 83.8 | 35.0 / 47.2 | 63.7 | 62.1 |

Table 8: **Comparison with Self-Cosistency.**

| Models | Methods | MuSR-ta | MuSR-op | GPQA | MATH500 | DROP | HotpotQA | MMLU-PRO | Avg. |
|---|---|---|---|---|---|---|---|---|---|
| LLaMA-3.1-8B-It | Self-Consistency | 76.4 | 54.7 | 26.3 | 52 | 73.0 / 78.0 | 23.0 / 29.8 | 45.1 | 50.1 |
| | ReFeri | **79.6** | **57.8** | **35.4** | 51.2 | 69.4 / 75.7 | 25.0 / 33.2 | 45.1 | 51.9 |
| | + Borda-vote(p=1) | 78.4 | 57.8 | 30.3 | 53.2 | **74.0 / 79.7** | **26.8 / 34.9** | **46.4** | **52.4** |
| GPT-4o-mini | Self-Consistency | 84.8 | 60.6 | 43.4 | **79.6** | 80.0 / 85.3 | 35.6 / 46.8 | 65.3 | 64.2 |
| | ReFeri | 82.8 | **61.3** | 41.9 | 77.8 | 79.2 / 84.9 | 36.2 / 48.0 | 64.9 | 63.4 |
| | + Borda-vote(p=1) | **86.8** | 60.9 | **44.4** | 79.4 | **80.4 / 86.0** | **36.8 / 48.3** | **65.7** | **64.9** |
| GPT-4o | Self-Consistency | 86.8 | 71.5 | 50.5 | 80 | 84.4 / 91.6 | 45.6 / 60.5 | **76.4** | 70.7 |
| | ReFeri | **90.4** | **71.9** | **51.5** | 77.8 | 83.6 / 91.1 | 47.0 / 60.7 | 75.4 | 71.1 |
| | + Borda-vote(p=1) | 88.8 | 71.5 | 49.5 | **80.2** | **84.8 / 92.0** | **48.0 / 61.9** | 76.2 | **71.3** |

from the selection prompt and using them only for scoring. Furthermore, LLM-as-Judge does not perform reliably on more complex tasks (*e.g.*, GPQA showing a noticeable accuracy degradation compared to other methods). These results emphasize that naively incorporating a few examples into prompts does not guarantee consistent gains, and that ReFeri is more robust and scalable. Finally, we note that the application of prompt-based approach could be limited due to inherent input context-window length. For reference, we also report an oracle upper bound in Table 5. This represents the accuracy achieved when selecting the optimal response from $K$ samples. This serves purely as a ceiling to describe in context how close each method approaches the maximum achievable performance. Although the gap with this ceiling is still noticeable, this highlights meaningful room for future improvements.

## B.2 ADDITIONAL COMPARISON WITH SELF-CONSISTENCY METHOD

As denoted in Section 1, self-consistency (majority voting) has an inherent limitation: it is only applicable when answers can be easily extracted and normalized for voting. For example, it is hard to be applied for open-ended text generation. For this reason, rather than the original self-consistency (Wang et al., 2023b), we mainly consider Universal Self-Consistency (USC) (Chen et al., 2023) as a baseline, which uses prompt-based evaluation to select among free-form outputs. In fact, HotpotQA is a representative case where standard self-consistency cannot be reliably applied, since mapping free-form outputs into consistent discrete answer categories is non-trivial.

Although standard Self-Consistency is difficult to apply in open-ended tasks, we conducted additional evaluations using standard self-consistency. To enable it on HotpotQA, we identified overlapping shared spans among free-form responses, counted how many responses contain each span, and used these aggregated counts as voting scores. In addition, inspired by the weighted voting scheme in Self-Certainty (Kang et al., 2025), we experimented with a Borda-voting method that uses the ReFeri metric as the weight, fixing the parameter $p = 1$ to avoid introducing new hyperparameters. Namely, the standard Self-Consistency selects the answer with the highest vote,

$$k_{\mathrm{sc}}^{\star} = \arg \max_{k \in \{1, \ldots, K\}} \sum_{j=1}^{K} \mathbf{1}[r_j = r_k],$$

in which each candidate contributes the same weight to one. The Borda vote generalizes this formula by replacing uniform unit weights as follows rank-based weights derived from the ReFeri score. Let rank($k$) denote the rank of candidate $r_k$ based on its ReFeri score. The corresponding borda weight is calculated as follows:

$$w_k = (K - \mathrm{rank}(k) + 1)^p$$

The final selection score of candidate $k$ is obtained by summing the weights of all candidates who make the same prediction:

$$k^\star_{\text{Borda}} = \arg \max_{k \in \{1,\dots,K\}} \sum_{j=1}^{K} w_k * \mathbf{1}[r_j = r_k],$$

The results are presented in the Table 8. Overall, ReFeri continues to outperform self-consistency on average. For instance, on MuSR-ta with LLaMA, ReFeri shows substantially higher performance (e.g., 76.4 vs. 79.6). More importantly, these methods are complementary: applying Borda voting with ReFeri yields notable improvements over conventional self-consistency, particularly on LLaMA.

### B.3 APPLICATION REFERI TO LLM PERSONALIZATION

Table 9: **LLM personalization.** Evaluation results on LaMP-4 and LaMP-5 using GPT-4o-mini as generator. *Vanilla* uses no history, while *Few-shot RAG* retrieves user history via BM25.

| Methods | LaMP-4 | | LaMP-5 | |
|---|---|---|---|---|
| | Rouge-1 | Rouge-L | Rouge-1 | Rouge-L |
| Vanilla | 0.120 | 0.106 | 0.421 | 0.332 |
| Few-shot RAG | 0.138 | 0.123 | 0.451 | 0.366 |
| ReFeri (Ours) | **0.143** | **0.128** | **0.470** | **0.394** |

We further apply ReFeri for *LLM personalization* to evaluate its broader applicability and more challenging open-ended tasks. The goal of LLM personalization is steering LLMs' responses towards the individual users, which becomes progressively important (Salemi et al., 2024; Tan et al., 2024; Kim & Yang, 2025). One representative baseline for LLM personalization is few-shot retrieval-augmented generation (RAG) that retrieved the user's previous data relevant to the given test query, and hence it's natural to apply ReFeri. Specifically, we evaluate on two tasks in *LaMP* benchmark (Salemi et al., 2024), LaMP-4 (personalized news headline generation) and LaMP-5 (personalized scholarly title generation), and use GPT-4o-mini as generation LLM. We generate $K = 5$ candidate responses with a temperature of 1.0 as same as Table 1. Building on the outputs generated through above pipeline, we apply our ReFeri method to select the most likely response among the five candidates for each input.

*Vanilla* baseline directly answers to query without external context, while the *Few-shot RAG* baseline augments input prompt with $N = 3$ examples retrieved via BM25 (Robertson et al., 2009) from the user's history. Following (Salemi et al., 2024), we evaluate all responses against gold references using ROUGE-1 and ROUGE-L. The average of all $K$ responses is reported for the baselines, and results with the selected response is reported for ReFeri, respectively. As shown in Table 9, ReFeri consistently outperforms both baselines across LaMP-4 and LaMP-5. Notably, it improves ROUGE-L from 0.366 to 0.394 on LaMP-5, and from 0.138 to 0.143 on LaMP-4. This result demonstrates the applicability of ReFeri beyond traditional reasoning tasks—to open-ended, user-specific scenarios.

### B.4 APPLICATION REFERI TO ZERO-SHOT RESPONSE

As shown in Table 1, Zero-shot CoT often achieves higher accuracy than Few-shot CoT, reflecting the intrinsic knowledge of the model. However, as described in Eq. 3, ReFeri is also applicable to selecting reasoning paths of Zero-shot CoT, although we primarily apply it to Few-shot CoT since it usually yields better reasoning paths (Table 1). With the experiments in Table 10, we verify that applying ReFeri to Zero-shot CoT yields improvements. These results further suggest that the few-shot exemplars in ReFeri mainly function as a post-hoc validation pipeline, rather than as generation guidance as in conventional Few-shot CoT. Also, this effectiveness of ReFeri under decoupling between generation and selection suggests a robust alternative to conventional few-shot prompting strategies, particularly in settings where few-shot examples are ineffective with LLMs.

### B.5 ROBUSTNESS TO SAMPLING STOCHASTICITY

Table 10: **Performance comparison between Zero-shot and ReFeri under zero-shot setting.**

| Models | Methods | MATH500 | GPQA | MuSR-ta |
|---|---|---|---|---|
| GPT-4o-mini | Zero-shot | 76.4 | 43.0 | 56.2 |
| | ReFeri | 78.2 | 43.9 | 58.8 |
| GPT-4o | Zero-shot | 77.5 | 48.8 | 66.6 |
| | ReFeri | 80.8 | 54.0 | 69.6 |
| LLaMA-3.1-8B-It | Zero-shot | 44.2 | 21.6 | 39.6 |
| | ReFeri | 50.8 | 24.2 | 41.2 |

Table 11: **Robustness to sampling stochasticity.** The overall results now yielded a total of three independent trial results, including two additional runs in the original Table 1.

| Models | Methods | MuSR-ta (Acc.) | MuSR-op (Acc.) | GPQA (Acc.) | MATH500 (Acc.) | DROP (EM / F1) | HotpotQA (EM / F1) | MMLU-PRO (Acc.) | Avg. |
|---|---|---|---|---|---|---|---|---|---|
| LLaMA-3.1-8B | Zero-shot CoT | 43.3 ±1.5 | 51.9 ±1.2 | 20.9 ±0.7 | 42.9 ±1.3 | 59.7 ±0.7 / 65.9 ±0.5 | 15.7 ±0.6 / 21.7 ±0.5 | 40.1 ±0.4 | 39.2 ±0.2 |
| | Few-shot CoT | 63.6 ±1.3 | 53.4 ±0.8 | 22.9 ±1.4 | 42.4 ±0.5 | 61.1 ±0.7 / 66.5 ±0.9 | 19.3 ±0.3 / 25.4 ±0.4 | 39.2 ±0.4 | 43.1 ±0.5 |
| | LEAP | 64.7 ±3.9 | 53.3 ±5.1 | 26.8 ±1.8 | 41.9 ±0.5 | 57.0 ±1.1 / 62.6 ±1.3 | 19.9 ±0.2 / 26.7 ±0.1 | 36.5 ±0.7 | 42.9 ±1.2 |
| | USC | 68.4 ±3.2 | 53.5 ±1.7 | 28.8 ±2.0 | 48.5 ±1.2 | 68.6 ±0.9 / 74.3 ±1.3 | 25.5 ±1.1 / 33.4 ±1.0 | 45.8 ±0.4 | 48.4 ±0.4 |
| | CoT-WP | 70.7 ±2.1 | 54.1 ±1.8 | 28.5 ±1.0 | 48.1 ±1.7 | 71.0 ±0.5 / 75.1 ±0.6 | 25.7 ±0.1 / 33.2 ±0.4 | 45.4 ±0.7 | 49.1 ±0.7 |
| | Self-Certainty | 74.3 ±1.6 | 55.9 ±0.6 | 30.1 ±3.2 | 50.7 ±2.3 | 70.8 ±0.9 / 76.2 ±0.7 | 25.2 ±0.5 / 32.9 ±1.0 | 44.5 ±0.8 | 50.2 ±0.9 |
| | ReFeri | 75.5 ±3.8 | 57.5 ±0.5 | 32.3 ±2.7 | 50.2 ±1.6 | 70.9 ±1.6 / 76.3 ±0.5 | 25.7 ±0.7 / 33.6 ±0.6 | 44.7 ±0.5 | 51.0 ±0.9 |
| GPT-4o-mini | Zero-shot CoT | 57.6 ±1.3 | 58.9 ±2.7 | 41.8 ±1.0 | 75.8 ±0.6 | 77.2 ±0.7 / 85.1 ±0.8 | 31.5 ±0.4 / 41.6 ±0.4 | 63.0 ±0.1 | 58.0 ±0.1 |
| | Few-shot CoT | 77.5 ±0.5 | 60.3 ±0.8 | 42.4 ±1.1 | 74.7 ±0.7 | 76.5 ±0.2 / 82.9 ±0.2 | 33.6 ±0.3 / 44.9 ±0.2 | 63.0 ±0.2 | 61.1 ±0.2 |
| | LEAP | 74.9 ±3.2 | 60.3 ±3.2 | 43.6 ±0.3 | 74.5 ±0.1 | 75.4 ±0.4 / 82.5 ±0.5 | 33.3 ±0.4 / 44.5 ±0.5 | 63.1 ±0.2 | 60.7 ±0.1 |
| | USC | 76.5 ±2.5 | 60.5 ±1.0 | 44.6 ±1.2 | 76.4 ±1.2 | 78.8 ±1.7 / 85.0 ±1.0 | 35.1 ±0.2 / 47.0 ±0.3 | 64.2 ±0.5 | 62.3 ±0.5 |
| | CoT-WP | 79.3 ±1.3 | 58.5 ±2.6 | 41.9 ±0.9 | 77.0 ±0.7 | 76.9 ±0.6 / 82.7 ±0.6 | 34.7 ±0.9 / 45.9 ±0.9 | 64.6 ±0.4 | 61.8 ±0.5 |
| | Self-Certainty | 81.7 ±1.6 | 60.3 ±0.8 | 41.6 ±2.8 | 76.8 ±1.0 | 76.8 ±0.2 / 83.2 ±0.4 | 34.7 ±0.1 / 45.9 ±0.4 | 63.9 ±0.8 | 62.2 ±0.2 |
| | ReFeri | 83.1 ±0.2 | 62.0 ±0.6 | 44.6 ±2.6 | 78.2 ±0.4 | 79.1 ±0.4 / 84.6 ±0.7 | 35.7 ±0.4 / 47.4 ±0.6 | 64.9 ±0.4 | 63.9 ±0.5 |
| GPT-4o | Zero-shot CoT | 67.5 ±0.8 | 62.1 ±0.4 | 49.5 ±0.8 | 77.1 ±0.6 | 74.2 ±0.8 / 84.9 ±0.4 | 37.8 ±0.3 / 50.3 ±0.4 | 74.0 ±0.2 | 63.2 ±0.2 |
| | Few-shot CoT | 87.2 ±0.6 | 69.6 ±0.6 | 47.3 ±1.7 | 75.5 ±0.1 | 80.4 ±0.2 / 89.0 ±0.2 | 44.9 ±0.4 / 58.6 ±0.3 | 73.7 ±0.2 | 68.4 ±0.3 |
| | LEAP | 88.1 ±1.6 | 68.0 ±1.2 | 47.8 ±2.8 | 75.2 ±0.4 | 81.0 ±0.5 / 89.4 ±0.4 | 44.5 ±0.6 / 57.8 ±0.5 | 73.9 ±0.2 | 68.4 ±0.4 |
| | USC | 85.9 ±1.9 | 71.5 ±0.7 | 48.2 ±1.6 | 77.3 ±0.3 | 81.8 ±0.3 / 90.2 ±0.1 | 45.9 ±0.3 / 60.2 ±0.5 | 74.9 ±0.5 | 69.4 ±0.5 |
| | CoT-WP | 88.3 ±0.2 | 67.5 ±1.4 | 49.5 ±1.8 | 78.1 ±0.6 | 83.3 ±0.1 / 90.5 ±0.8 | 46.3 ±0.9 / 59.6 ±0.9 | 74.4 ±0.3 | 69.6 ±0.0 |
| | Self-Certainty | 88.9 ±0.5 | 71.4 ±2.3 | 50.3 ±0.6 | 77.7 ±0.4 | 81.1 ±0.8 / 89.4 ±0.3 | 44.0 ±0.4 / 57.8 ±0.1 | 74.6 ±0.4 | 69.7 ±0.4 |
| | ReFeri | 90.8 ±0.4 | 73.7 ±2.2 | 51.3 ±0.3 | 78.5 ±0.6 | 83.6 ±0.2 / 90.9 ±0.6 | 46.9 ±0.4 / 60.9 ±0.4 | 75.4 ±0.3 | 71.5 ±0.4 |

To evaluate the stability of ReFeri under sampling variability, we conduct multiple run experiments against Table 1 in which each run samples a new set of candidate responses from a generation model. By design, baseline methods (e.g., USC, CoT-WP) and ReFeri work deterministically by sharing exactly the same set of fixed candidate responses, and the reported results has no randomness. However, since the candidates themselves are subject to sampling stochasticity, we perform multiple runs to evaluate the consistency of the performance gains with newly added baseline Self-Certainty (Kang et al., 2025).

Self-Certainty uses predictive distributions in practice to estimate the uncertainty of responses, which can be viewed as a purely forward approach. This is conceptually similar to CoT-WP, but Self-Certainty uses entropy-based uncertainty signals (KL-Divergence) instead of log probability gaps. We further incorporate this forward mechanism into the evaluation set to measure the contribution of the backward term more clearly.

The results of multiple runs with Self-Certainty are presented in Table 11. While Self-Certainty is a strong and competitive baseline, ReFeri consistently outperforms. For example, on the GPT-4o-mini model, ReFeri achieves an average accuracy of $64.2 \pm 0.4$, exceeding the $62.3 \pm 0.2$ compared to the Self-Certainty baseline. Similarly, on the GPT-4o, ReFeri also outperformed on $71.6 \pm 0.3$ compared to $69.7 \pm 0.5$. For a fair comparison, and to remain consistent with our evaluation setup, we used LLaMA-3.1-8B-Instruct to compute the self-certainty scores, since log probabilities for closed-source models such as GPT-4o are not directly accessible.

Interestingly, while Self-Certainty provides notable benefits for smaller open-source models (e.g., LLaMA), which often produce more verbose or stylistically variable responses, its effectiveness diminishes for stronger models such as GPT-4o-mini and GPT-4o. We conjecture this is because the sampled responses In these large models are uniformly high-quality and have very similar predictive distributions, making entropy-based uncertainty insufficient for distinguishing subtle differences. In contrast, ReFeri remains effective because it leverages few-shot demonstrations during validation,

Table 12: **Results of GPT-4o-mini across different few-shot examples and ReFeri.**

| Methods | MATH500 | GPQA | MuSR-ta |
|---|---|---|---|
| Few-shot 1 | 75.2 | 41.3 | 77.0 |
| ReFeri 1 | 77.8 | 41.9 | 82.8 |
| Few-shot 2 | 74.5 | 41.5 | 57.8 |
| ReFeri 2 | 79.0 | 43.4 | 59.2 |
| Few-shot 3 | 75.0 | 38.9 | 60.1 |
| ReFeri 3 | 77.8 | 41.9 | 62.8 |

not just generation. Few-shot examples offer a compact way to inject human prior knowledge, and ReFeri uses this external information to complement internal model confidence. As a result, unlike forward-only approaches that depend solely on the model's intrinsic distributional signals, ReFeri incorporates external human insights and generalizes more reliably across models and tasks. These observations suggest that while forward-only metrics become unreliable in realistic scenarios where powerful LLMs produce uniformly confident outputs, ReFeri maintains robustness by integrating complementary backward information grounded in few-shot demonstrations.

### B.6 MORE RESULTS WITH DIFFERENT FEW-SHOT EXAMPLES

In addition to the results reported in Section 3.3, we provide extended experiments in Table 12 including MuSR-ta benchmark. Interestingly, MuSR-ta once again highlights the importance of example quality; when synthesizing new data according to (Sprague et al., 2024) to use as few-shot examples, baseline accuracy significantly degrades. Nevertheless, ReFeri demonstrates consistent performance improvements and confirming the robustness.

Table 13: **Performance of ReFeri in weak few-shot settings.**

| Models | Methods | MATH500 | GPQA | MuSR-ta |
|---|---|---|---|---|
| GPT-4o-mini | Few-shot | 73.7 | 41.1 | 57.5 |
| | ReFeri | 76.4 | 43.4 | 58.0 |
| GPT-4o | Few-shot | 75.2 | 46.1 | 71.0 |
| | ReFeri | 79.0 | 47.0 | 75.6 |
| LLaMA-3.1-8B-It | Few-shot | 39.5 | 26.6 | 38.7 |
| | ReFeri | 47.6 | 33.3 | 41.6 |

We believe that constructing accurate few-shot examples is a minimal effort that one should invest to guide LLMs (even humans) toward a proper behavior for a target task. Still, to evaluate robustness, we conducted experiments using intentionally synthesized "weak few-shot" (Table 13) by GPT-4o-mini. Even under these weaker conditions, ReFeri continues to improve performance relative to the Few-shot CoT, confirming that verification remains effective even with suboptimal examples and can provide meaningful gains in more practical, less curated scenarios.

Table 14: **Judgment scores (1–10) by GPT-4o for weak fewshot quality.**

| Judge by GPT-4o (1–10) | MATH500 | GPQA | MuSR-ta |
|---|---|---|---|
| Few-shot | 8 | 8 | 8 |
| Low quality | 3 | 5 | 4 |

To verify the degradation, we asked GPT-4o to evaluate the quality of the original set and the weak example set. The evaluation was conducted in random order, and information about each set was not provided to avoid bias. As shown in Table 14, the weak set consistently received significantly lower scores (3–5 points) compared to the original examples (8 points). This further demonstrates that ReFeri maintains its effectiveness even when the quality of the provided examples is low. See list 10-11 for exact prompts used in the generation and evaluation assessment.

Table 15: **Ablation on generation/evaluation prompts.**

| Gen Prompt | Eval Prompt | MATH500 | | GPQA | | MuSR-ta | |
|---|---|---|---|---|---|---|---|
| | | Few-shot | ReFeri | Few-shot | ReFeri | Few-shot | ReFeri |
| **Orig** | Orig | 75.2 | **77.8** | 41.3 | **41.9** | 77.0 | **82.8** |
| | Plan | 75.2 | **78.0** | 41.3 | **42.4** | 77.0 | **82.8** |
| | Role | 75.2 | **77.8** | 41.3 | **41.9** | 77.0 | **82.4** |
| **Plan** | Plan | 74.6 | **78.2** | 42.6 | **47.5** | 77.0 | **82.4** |
| | Orig | 74.6 | **78.4** | 42.6 | **47.5** | 77.0 | **82.4** |
| **Role** | Role | 74.5 | **78.2** | 43.5 | **47.5** | 75.8 | **81.6** |
| | Orig | 74.5 | **78.2** | 43.5 | **47.0** | 75.8 | **81.6** |

Table 16: **Full results with different estimation models across three benchmarks.**

| (a) MATH500 | | | | |
|---|---|---|---|---|
| Estimation | GPT-4o-mini | GPT-4o | LLaMA-3.1-8B-It | Avg |
| LLaMA-3.2-1B | 78.0 | 77.6 | 51.4 | 69.0 |
| LLaMA-3.1-8B | 77.8 | 77.8 | 51.2 | 68.9 |
| Qwen-2.5-7B | 78.8 | 79.2 | 52.0 | 70.0 |
| LLaMA-3.1-70B | 77.8 | 77.6 | 53.6 | 69.7 |

| (b) GPQA | | | | |
|---|---|---|---|---|
| Estimation | GPT-4o-mini | GPT-4o | LLaMA-3.1-8B-It | Avg |
| LLaMA-3.2-1B | 43.9 | 50.5 | 33.8 | 42.7 |
| LLaMA-3.1-8B | 41.9 | 51.5 | 35.4 | 42.9 |
| Qwen-2.5-7B | 41.4 | 50.5 | 34.3 | 42.1 |
| LLaMA-3.1-70B | 42.4 | 53.5 | 34.8 | 43.6 |

| (c) MuSR-ta | | | | |
|---|---|---|---|---|
| Estimation | GPT-4o-mini | GPT-4o | LLaMA-3.1-8B-It | Avg |
| LLaMA-3.2-1B | 83.2 | 90.8 | 80.0 | 84.7 |
| LLaMA-3.1-8B | 82.8 | 90.4 | 79.6 | 84.3 |
| Qwen-2.5-7B | 82.0 | 90.8 | 81.6 | 84.8 |
| LLaMA-3.1-70B | 83.6 | 91.2 | 81.6 | 85.5 |

## B.7 MORE RESULTS ON GENERATION/EVALUATION PROMPTS

In addition to the prompt style (see 12, 13) ablation study reported in Section 3.3, Table 15 extends the results to include MuSR-ta. As mentioned above, ReFeri demonstrates stable performance across various combinations of generation and evaluation prompts (orig, plan, role), indicating robustness to changes in prompt style. The accuracy of responses generated by Few-shot CoT varies depending on the generation style, but ReFeri consistently shows improved performance across all configurations.

## B.8 FULL RESULTS WITH DIFFERENT ESTIMATION MODELS

Table 17: **Computational cost.** Evaluation cost of GPT-4o-mini. Costs are measured in actual processing time (seconds) per instance on a single GPU using the same model configuration.

| Size | Methods | MATH500 | GPQA | MuSR-ta |
|---|---|---|---|---|
| | | (Acc. / Time) | (Acc. / Time) | (Acc. / Time) |
| 1B | USC | 75.0 / 0.6 | 44.9 / 0.1 | 75.6 / 0.7 |
| | CoT-WP | 76.0 / 1.5 | 43.4 / 2.0 | 77.6 / 5.0 |
| | ReFeri(Full) | 78.0 / 9.6 | 44.9 / 12.6 | 83.2 / 21.3 |
| | ReFeri | 78.0 / 3.0 | 43.9 / 4.0 | 83.2 / 8.0 |
| 8B | USC | 77.8 / 3.7 | 46.0 / 3.7 | 74.4 / 3.9 |
| | CoT-WP | 77.8 / 8.3 | 42.4 / 11.0 | 78.8 / 25.6 |
| | ReFeri | 77.8 / 16.6 | 41.9 / 22.1 | 82.8 / 41.8 |

Table 16 provides full results for all estimation model combinations of MATH500, GPQA and MuSR-ta. This complements the average performance across different generation LLMs (GPT-4o-mini, GPT-4o, and LLaMA3.1-8B) shown in Figure 4. Across all three tasks, ReFeri shows consistent performance gains regardless of the estimation model used, emphasizing its robustness. There are

some model-specific trends; for example, smaller models (LLaMA-3.2-1B) perform competitively on (relatively) simple tasks like MATH500, as discussed in Section 3.2.

Moreover, we provide further results by including MuSR-ta in a cost-accuracy analysis (Table 17), which complements the discussion in Sec. 3.3. On this benchmark, name with 1B estimator achieves 83.2% accuracy while requiring only 8s per query, but clearly outperforms the robust 8B CoT-WP baseline, which achieves 78.8% but consumes more than three times the latency (25.6s). This result illustrates that ReFeri with a smaller estimator can still effectively utilizes a few-shot examples to provide robust validation at a much lower cost, making it particularly attractive for scenarios where latency and resource budgets are critical.

While the lightweight approximation offers significant computational advantages, replacing the entire Bayesian term in Eq. 5 with the single most relevant example implies a theoretical simplification. This reduction may suggest a departure from the ostensibly rigorous Bayesian rule.

However, previous work on in-context learning has observed that the relative contributions of examples are highly uneven, and that the most relevant examples often account for a disproportionately large proportion of useful signals (Wang et al., 2024a; Li et al., 2023; Liu et al., 2021). From this perspective, the lightweight version does not replace the conceptual role of the entire backward component, but only provides a tractable replacement. Our experiments support this interpretation. As shown in Table 18, the performance gap between the full backward computation and the lightweight version ReFeri that we suggests is negligible across benchmarks, while the lightweight variant reduces computation substantially.

Furthermore, the modularity that separates generation from estimation allows our method to maintain its theoretical validity, even with smaller estimation models. This not only avoids the collapse of Bayesian interpretation, but also provides practical efficiency benefits. As shown in Table 17, even with full backward computation, ReFeri Full (1B) achieves higher accuracy than CoT-WP using 8B estimators, despite requiring similar or lower computation. For example, in MuSR-ta, ReFeri Full (1B) achieves 83.2% accuracy with a cost per query of 21.3s, while CoT-WP (8B) achieves 78.8% accuracy with 25.6s. On the other hand, CoT-WP experiences noticeable accuracy degradation when reducing the estimator from 8B to 1B, while ReFeri maintains stable performance across model sizes.

These results indicate that the lightweight approximation does not collapse the theoretical framework. ReFeri still maintains a conceptual Bayesian structure, and maintains the benefits of backward consistency. We believe that this effectiveness, even when the generation and estimation models are not aligned, is a key strength of our approach. This design choice makes ReFeri broadly applicable and can be easily integrated to existing pipelines.

## B.9 ADDITIONAL ABLATION

Table 18: **Additional ablation study on GPT-4o-mini**

| Methods | MuSR-ta (Acc.) | MuSR-op (Acc.) | GPQA (Acc.) | MATH500 (Acc.) | DROP (EM / F1) | HotpotQA (EM / F1) | MMLU-PRO (Acc.) | Avg. |
|---|---|---|---|---|---|---|---|---|
| No replace (full) | **82.8** | 60.2 | 42.4 | **78.0** | 78.4 / 84.2 | **36.2 / 48.0** | **65.0** | 63.3 |
| No replace | 82.4 | 60.2 | **42.9** | 77.6 | 78.4 / 84.1 | 35.8 / 47.6 | 64.7 | 63.1 |
| ReFeri (Full) | **82.8** | **61.3** | 42.4 | 77.8 | **79.6 / 85.3** | 35.8 / 47.9 | **65.0** | **63.5** |
| ReFeri | **82.8** | **61.3** | 41.9 | 77.8 | 79.2 / 84.9 | **36.2 / 48.0** | 64.9 | 63.4 |

Here, we conduct the additional experiments to provide comprehensive ablation study for ReFeri. We first evaluate the *Full* variant (Eq. 8), which generally achieves the strongest results across benchmarks (Table 18). This is expected, as using the complete set of examples provides the most faithful estimate of backward consistency. However, as discussed in Sec. 2.2, the computational overhead increases linearly with the number of few-shot examples, which renders the *Full* variant less appealing for large-scale or resource-constrained scenarios.

To further analyze this trade-off, we examine the effectiveness of the proposed *prompt replacement* (Eq. 7) for better estimation of backward score. To this end, we consider a simplified variant of our backward score, termed *No replace*, where each few-shot example $\mathbf{x}_i = (q_i, a_i)$ is evaluated

in a one-shot manner using the test query $\widetilde{q}$ and the candidate response $r_k$ as additional context. Specifically, this variant modifies the backward score in Eq. 8 by replacing the leave-one-out prompt $\widetilde{\mathbf{X}}_i$ with a single pair $\mathbf{y}_k = (\widetilde{q}, r_k)$:

$$S'_{\text{Back}}(r_k) := \log P(\mathbf{X} \mid \mathbf{y}_k) - \log P(\mathbf{X}) = \sum_{i=1}^{N} \left[ \log P(a_i \mid q_i, \widetilde{q}, r_k) - \log P(a_i \mid q_i) \right], \quad (12)$$

We note that, as in our main method, a cost-efficient variant can be obtained by incorporating the $i^\dagger$ selection strategy (Eq. 9), which adaptively chooses the most relevant exemplar to the test query.

$$S'_{\text{Back}}(r_k) := \log P(a_{i\dagger} \mid q_{i\dagger}, \widetilde{q}, r_k) - \log P(a_{i\dagger} \mid q_{i\dagger}), \quad (13)$$

This formulation can be interpreted as the most straight-forward implementation of backward score (see Eq. 5) under the assumption of mutual independence between few-shot examples. As shown in Table 18, the accuracy under *No replace* is consistently less or equal than ReFeri (6 of 7). We attribute this to the fact that using full leave-one-out prompts better reflects the consistency of $\mathbf{y}_k$ with the original in-context reasoning trajectory. Nonetheless, *No replace* could serve as a practical alternative that trades off a small performance drop with the greater simplicity.

Table 19: **Additional ablation on the interpretive role of the backward score.** Using each candidate as a one-shot demonstration, we evaluate whether the backward score correlates with the ability to reconstruct the few-shot.

| Task | 1 | 2 | 3 | 4 | 5 |
|------|------|------|------|------|------|
| MATH500 | 87.0 | 85.4 | 85.2 | 84.2 | 84.5 |
| MuSR-ta | 37.2 | 37.6 | 33.7 | 30.3 | 29.9 |

To further examine the interpretability of the backward score, we conducted an additional experiments. Specifically, we take the generated outputs from LLaMA-3.1-8B-Instruct, ranked them by their backward score, and then used each output as a one-shot demonstration to solve the original few-shot queries. For a more intuitive understanding, we used the *No-replace* backward score (Eq. 13). This term is a simplified variant where each few-shot examples $\mathbf{x}_i$ is evaluated in a one-shot manner using test query $\widetilde{q}$ and the candidate response $r_k$ as condition. By utilizing the *No-replace* backward score, we isolate the specific impact of the candidate response $r_k$ on measure a single few-shot example $\mathbf{x}_i$, thereby eliminating the confounding factor of other few-shot demonstrations.

As shown in Table 19, the results demonstrate a correlation where responses with higher backward score is better capable of guiding the model to answer the original questions, confirming that they capture the underlying reasoning **archetype** of the few-shot examples. In other words, the backward term is not merely a heuristic, but in practice reflects whether a candidate serves as a strong archetype for the task distribution.

## B.10 Robustness to estimation model calibration

While main experiments used a fixed temperature of $T = 1.0$ for the estimation model, natural question is how sensitive ReFeri to the calibration of the estimator. To investigate, we conducted a series of experiments altering the model calibration via temperature scaling, adjusting the logit value to $T \in \{0.5, 1.0(\text{default}), 1.5, 2.0\}$. As shown in the Table 20, indicate that ReFeri remains remarkably robust across temperatures, showing only minimal variation even under substantial overconfidence or underconfidence.

This robustness is further demonstrated through cross-model and cross-scale evaluations. As shown in Figure 4 and Appendix B.8 (Table 16), ReFeri using the 1B estimation model maintains stable performance across various tasks and calibration environments, whereas the likelihood-based baseline CoT-WP exhibits significantly greater variability. These observations suggest that ReFeri does not heavily rely on the precise calibration of the estimation model. The backward term provides an additional signal that helps compensate for calibration drift by capturing explanatory alignment with the few-shot examples rather than relying solely on raw likelihood peaks.

Table 20: **Robustness to estimation model calibration.** We report the performance of ReFeri across different temperature scaling factors $T \in \{0.5, 1.0, 1.5, 2.0\}$ applied to the estimation model.

| Models | Temp | MuSR-ta | MuSR-op | GPQA | MATH500 | DROP | HotpotQA | Avg. |
|---|---|---|---|---|---|---|---|---|
| LLaMA-3.1-8B-It | 0.5 | **79.6** | 57.0 | 33.3 | 50.6 | 69.8 / 75.9 | 24.6 / 33.0 | 52.5 |
| | 1 | **79.6** | **57.8** | **35.4** | 51.2 | 69.4 / 75.7 | 25.0 / **33.2** | **53.1** |
| | 1.5 | 78.8 | 57.0 | 34.8 | **52.0** | 70.2 / 76.1 | 25.2 / **33.2** | 53.0 |
| | 2 | 78.0 | 56.2 | 34.3 | 51.8 | **71.0 / 77.1** | 24.6 / 32.7 | 52.7 |
| GPT-4o-mini | 0.5 | 82.0 | 59.4 | **43.9** | 78.0 | 78.0 / 83.8 | 35.8 /47.1 | 62.9 |
| | 1 | 82.8 | **61.3** | 41.9 | 77.8 | **79.2 / 84.9** | 36.2 / 48.0 | 63.2 |
| | 1.5 | **83.6** | 60.2 | 43.4 | 78.0 | 78.0 / 84.7 | **36.4 / 48.3** | **63.3** |
| | 2 | 83.2 | 59.0 | 41.4 | **78.4** | 77.4 / 84.1 | 35.2 / 47.6 | 62.4 |
| GPT-4o | 0.5 | 90.4 | 71.1 | **52.0** | 77.4 | **84.0 / 91.3** | **47.6 / 60.7** | **70.4** |
| | 1 | 90.4 | 71.9 | 51.5 | 77.8 | 83.6 / 91.1 | 47.0 / **60.7** | **70.4** |
| | 1.5 | **91.2** | 71.1 | 50.5 | 77.6 | 83.4 / 91.1 | 45.0 / 59.2 | 69.8 |
| | 2 | 90.4 | **71.9** | 51.0 | 77.6 | 81.8 / 89.9 | 44.8 / 59.3 | 69.6 |

## C  USAGE OF AI ASSISTANTS

This paper used AI-based writing aids to improve sentence structure, correct grammar, and improve readability. These tools were applied only to language refinement and did not affect the advancement of technical content, research methodology, or experimental analysis. All scientific ideas, results, and conclusions were conceived and written entirely by researchers. The use of AI aids was limited to editorial purposes and did not impair the originality or intellectual contribution of the work.

## D  QUALITATIVE EXAMPLES

In this section, we present qualitative examples to further analyze the proposed ReFeri. For better readability, we only present the examples from MATH500, GPQA, and HotpotQA. All the responses are generated by GPT-4o-mini, and we use the ReFeri *(Full)* variant for illustration to provide the clearest comparisons.

### D.1  TOKEN LEVEL ANALYSIS

To better understand how ReFeri identifies high-quality response using given few-shot examples, we perform a token-level analysis of following backward consistency score (Eq. 8).

For a given test query $\widetilde{q}$, we divide the candidate responses into correct and incorrect groups using ground-truth labels, and calculate the difference in token level score between the two groups. When the backward scores for the tokens in the few-shot examples exhibit lower score in the correct group compared to incorrect one, the tokens are colored *red*. In other case, the tokens are colored *blue*. For visual clarity, we only highlight the top 60% of tokens based on the absolute difference in values. The remaining 40% remain uncolored. This visualization highlights the tokens that contributed the most to plausible candidate answers as determined by the backward consistency score. The value in parentheses is the ratio of tokens highlighted in red to the total number of tokens.

Lower token-level scores indicate higher validity in the model, so tokens highlighted in red can be interpreted as those where backward consistency most effectively distinguishes plausible response. For example, in MATH500 and GPQA, we found that numbers, symbols, final answer formatting, or next reasoning steps were often colored red to support the interpretability of the selection criteria for ReFeri. In HotpotQA can also see that red is dominant for most, but not all, meaningful words.

Few-shot examples for MATH500 (0.6455):

**Test-Query**:
```
Twelve 1 by 1 squares form a rectangle, as shown.  What is the total
area of the shaded region?  [asy]
size(4cm);defaultpen(linewidth(0.75));

// Filled portions
fill((0, 4)-(0, 0)-(2, 0)-cycle, gray(0.75));
fill((0, 4)-(3, 4)-(3, 0)-cycle, gray(0.75));

// grid
int j;
for (j = 0; j < 4; ++j) draw((j, 0)-(j, 4));
for (j = 0; j < 5; ++j) draw((0, j)-(3, j));

//diagonals
draw((0, 4)-(3, 0)); draw((0, 4)-(2, 0));
[/asy]
```

**Few-shot 1 Query**:
Kevin Kangaroo begins hopping on a number line at 0.  He wants to get
to 1, but he can hop only $\frac{1}{3}$ of the distance.  Each hop tires him out so
that he continues to hop $\frac{1}{3}$ of the remaining distance.  How far has he
hopped after five hops?  Express your answer as a common fraction.
**Few-shot 1 Response**:
Let's think step by step
Kevin hops $1/3$ of the remaining distance with every hop.
His first hop takes $1/3$ closer.
For his second hop, he has $2/3$ left to travel, so he hops forward
$(2/3)(1/3)$.
For his third hop, he has $(2/3)^2$ left to travel, so he hops forward
$(2/3)^2(1/3)$.
In general, Kevin hops forward $(2/3)^{k-1}(1/3)$ on his $k$th hop.
We want to find how far he has hopped after five hops.
This is a finite geometric series with first term $1/3$, common ratio
$2/3$, and five terms.
Thus, Kevin has hopped $\frac{\frac{1}{3}\left(1-\left(\frac{2}{3}\right)^5\right)}{1-\frac{2}{3}} = \boxed{\frac{211}{243}}$.
The answer is \frac{211}{243}}

**Few-shot 2 Query**:
What is the area of the region defined by the equation $x^2+y^2-7=4y-14x+3$?
**Few-shot 2 Response**:
Let's think step by step
We rewrite the equation as $x^2 + 14x + y^2 - 4y = 10$ and then
complete the square,
resulting in $(x+7)^2-49 + (y-2)^2-4=10$,
or $(x+7)^2+(y-2)^2=63$.
This is the equation of a circle with center $(-7, 2)$ and radius
$\sqrt{63},$
so the area of this region is $\pi r^2 = \boxed{63\pi}$.
The answer is 63\pi

Few-shot examples for MATH500 (0.6455):

**Few-shot 3 Query:**
If $x^2 + y^2 = 1$, what is the largest possible value of $|x| + |y|$?
**Few-shot 3 Response:**
Let's think step by step
If $(x,y)$ lies on the circle,
so does $(x,-y),$ $(-x,-y),$ and $(-x,-y),$ (which all give the same
value of $|x| + |y|$),
so we can assume that $x \ge 0$ and $y \ge 0.$Then $|x| + |y| = x + y.$
Squaring, we get \[(x + y)^2 = x^2 + 2xy + y^2 = 1 + 2xy.\]
Note that $(x - y)^2 \ge 0.$Expanding, we get $x^2 - 2xy + y^2 \ge 0,$
so $2xy \le x^2 + y^2 = 1.$ Hence,\[1 + 2xy \le 2,\]which means $x + y
\le \sqrt{2}.$Equality occurs when $x = y = \frac{1}{\sqrt{2}},$
so the maximum value of $|x| + |y|$ is $\boxed{\sqrt{2}}.$
The answer is \sqrt{2}

**Few-shot 4 Query:**
If $f(x) = \frac{ax+b}{cx+d}, abcd \neq 0$ and $f(f(x)) = x$ for all $x$ in the domain of $f$, what is
the value of $a + d$?
**Few-shot 4 Response:**
Let's think step by step
The condition $f(f(x))$ means that $f$ is the inverse of itself,
so its graph is symmetrical about the line $y = x$.
With a rational function of this form, we will have two asymptotes:
a vertical one at $x=-d/c$ if $cx+d$ does not divide $ax+b$,
and a horizontal one at $y=a/c$,
if we take the limit of $f(x)$ as $x$ goes to $\pm\infty$.
In order for $f$ to be its own inverse, the intersection of the
asymptotes must lie on the line $y=x$
so that it and its asymptotes reflect onto themselves.
This means that $-d/c=a/c$,
and therefore $-d=a$ and $a+d=\boxed{0}$.
The answer is 0

**Few-shot 5 Query:**
Expand $(2z^2 + 5z - 6)(3z^3 - 2z + 1)$.
**Few-shot 5 Response:**
Let's think step by step
$$\begin{array}{crrrrrr}
& & & 3z^3 & & -2z & + 1 & \\
\times & & & & 2z^2 & +5z & -6 \\
\cline{1-7}\rule{0pt}{0.17in}
& & & -18z^3 & & +12z & -6 & \\
& & +15z^4 & & -10z^2 & +5z & & \\
+ & & 6z^5 & & -4z^3 & +2z^2 & & & \\
\cline{1-7}\rule{0pt}{0.17in}
& 6z^5 & +15z^4 & -22z^3 & - & 8z^2 &+17z & -6 &
\end{array}$$ The answer is 6z^5+15z^4-22z^3-8z^2+17z-6

Few-shot examples for GPQA (0.6970):

**Test-Query:**
A chemist performs two reactions:

Reaction 1: (E)-oct-4-ene is treated with one equiv. of mCPBA,
followed by aqueous acid.

Reaction 2: (Z)-oct-4-ene is treated with one equiv. of mCPBA,
followed by aqueous acid.

Both reactions go to 100% completion. The chemist combines the
products of both reactions, and then runs the product mixture on both a
standard (achiral) reverse-phase HPLC column and a chiral HPLC column.

Assuming that the chromatographic resolution is as high as
theoretically possible in both cases, what does the chemist observe
in each of these chromatograms?

**Few-shot 1 Query:**
In a given population, 1 out of every 400 people has a cancer caused
by a completely recessive allele, b. Assuming the population is in
Hardy-Weinberg equilibrium, which of the following is the expected
proportion of individuals who carry the b allele but are not expected
to develop the cancer?
**Few-shot 1 Response:**
The expected proportion of individuals who carry the b allele but
are not expected to develop the cancer equals to the frequency of
heterozygous allele in the given population.
According to the Hardy-Weinberg equation $p^2 + 2pq + q^2 = 1$, where
p is the frequency of dominant allele frequency, q is the frequency
of recessive allele frequency, $p^2$ is the frequency of the homozygous
dominant allele, $q^2$ is the frequency of the recessive allele, and 2pq
is the frequency of the heterozygous allele.
Given that $q^2=1/400$, hence, q=0.05 and p=1-q=0.95.
The frequency of the heterozygous allele is 2pq=2*0.05*0.95=38/400.

**Few-shot 2 Query:**
A Fe pellet of 0.056 g is first dissolved in 10 mL of hydrobromic acid
HBr (0.1 M). The resulting solution is then titrated by KMnO4 (0.02 M).
How many equivalence points are there?
**Few-shot 2 Response:**
HBr reacts with Fe to produce Fe2+.
MnO4- initially reacts with Fe2+ followed by Br-.
There are two equivalence points at 25 ml and 35 ml.
In the beaker, the present species are Fe2+ and Br-.
In a titration involving two analytes, it's essential to identify which
reaction occurs first.
Given the redox nature of the titration and the reduction potentials:
E0 (Br2/Br-) = 1.09 V, E0 (MnO4-/Mn2+) = 1.49 V, and E0 (Fe3+/Fe2+) =
0.77 V.
With [Fe2+] determined as 0.1M, two reactions are considered.
Reaction 1: MnO4- reacts with 5Fe2+ and 8H+ to produce Mn2+, 5Fe3+, and
4H2O.
Reaction 2: 2MnO4- reacts with 10Br- and 16H+ to produce 2Mn2+ and 5Br2
with 8H2O as a byproduct.
MnO4- first reacts with Fe2+ in a 1:5 ratio, making the first
equivalence point at 10 ml.
Once Fe2+ is exhausted, MnO4- reacts with Br- in a 2:10 ratio, adding
another 25 ml for a total second equivalence point at 35 ml.

(cut)

---

Few-shot examples for GPQA (0.6970):

---

**Few-shot 3 Query:**
Consider a quantum mechanical system containing a particle of mass $m$ moving in an istropic three dimensional potential of the form $V(r) = 1/2m\omega^2 r^2$ corresponding to the acted force obeying Hooke's law. Here, $\omega$ is the angular frequency of oscillation and $r$ is the radial distance of the particle from the origin in spherical polar coordinate. What is the value of energy of the third excited state, and how many linearly independent eigenfunctions are possible for the same energy eigenvalue?

**Few-shot 3 Response:**
This problem is nothing but the three dimensional simple harmonic oscillator (SHO) problem.
The energy spectrum of three dimensional SHO is $E_n= (n+3/2)\hbar \omega$ where $n=0,1,2,3....$.
For third excited state n=3.
3+3/2=6/2+3/2=9/2.
Thus the corresponding energy is $(9/2)\hbar \omega$.
The degeneracy of the state is $g_n= (n+1)(n+2)/2$.
For n=3, degeneracy is $(3+1)*(3+2)/2=4*5/2=10$.

---

**Few-shot 4 Query:**
Your overhear two chemists talking to each other as they leave a synthetic organic chemistry lab. One asks the other "So, how did it go?" The second chemist replies, "Not well – my compounds are on top of each other." What is the second chemist most likely referring to?

**Few-shot 4 Response:**
"On top of each other" commonly refers to two compounds that have similar Rf values on chromatography (a common operation in synthetic chemistry).
Similar Rf values arise for compounds with similar polarities.

---

**Few-shot 5 Query:**
Mitochondria are semi-autonomous cellular organelles in charge of energy production. They encode for a part of their own translational machinery and respiratory complexes. Mitochondrial function is governed by over a thousand proteins imported from the cell, contributing to processes like the transport of proteins, ribosome biogenesis and translation regulation, respiratory oxidation, metabolism, and apoptotic signaling cascade. Mutations in the code for mitochondrial protein networks can cause numerous diseases in humans that are inherited through generations. Mutations of which of the mitochondrial proteins listed below are least likely to be genetically transmitted from a father to his children?

**Few-shot 5 Response:**
The colleague should know that mitochondria from fathers are rarely if ever, transmitted to their offspring.
Therefore, the protein encoded by the paternal mitochondrial genome will most likely not be passed down the generation.
NADH dehydrogenase 2 is the only one encoded by the mitochondrial genome from the MT-ND2 gene among the listed proteins.
Leigh's syndrome, lactic acidosis, and metabolic diseases are all linked to a mutation in the ND2 gene.
ATP binding cassette subfamily B member 8 (ABCB8) is a chromosome 7 encoded gene; Tu translation elongation factor, mitochondrial is chromosome 16 gene TUFM.
Translocase of inner mitochondrial membrane 17B is chromosome X coded gene TIMM17B.
There is no evidence that it is maternally imprinted; hence, daughters may inherit the father's gene copy in a 50:50 ratio.

---

---

Few-shot examples for HotpotQA (0.7883):

---

**Test-Query:**
H-63 is a highway that runs near the county seat of what Michigan county?.

---

**Few-shot 1 Query:**
What is the elevation range for the area that the eastern sector of the Colorado orogeny extends into?
**Few-shot 1 Response:**
's think step by step. The eastern sector of Colorado orogeny extends into the High Plains. High Plains rise in elevation from around 1,800 to 7,000 ft, so the answer is 1,800 to 7,000 ft. Answer 1,800 to 7,000 ft

---

**Few-shot 2 Query:**
Musician and satirist Allie Goertz wrote a song about the "The Simpsons" character Milhouse, who Matt Groening named after who?
**Few-shot 2 Response:**
's think step by step. Milhouse was named after U.S. president Richard Nixon, so the answer is Richard Nixon. Answer Richard Nixon

---

**Few-shot 3 Query:**
Which documentary is about Finnish rock groups, Adam Clayton Powell or The Saimaa Gesture?
**Few-shot 3 Response:**
's think step by step. Adam Clayton Powell (film) is a documentary about an African-American politician, not Finnish rock groups. So the documentary about Finnish rock groups must instead be The Saimaa Gesture. Answer The Saimaa Gesture

---

**Few-shot 4 Query:**
What profession does Nicholas Ray and Elia Kazan have in common?
**Few-shot 4 Response:**
's think step by step. Professions of Nicholas Ray are director, screenwriter, and actor. Professions of Elia Kazan are director, producer, screenwriter, and actor. So profession Nicholas Ray and Elia Kazan have in common is director, screenwriter, and actor. Answer director, screenwriter, actor

---

**Few-shot 5 Query:**
Which magazine was started first Arthur's Magazine or First for Women?
**Few-shot 5 Response:**
's think step by step. Arthur's Magazine was started in 1844. First for Women was started in 1989. 1844 (Arthur's Magazine) < 1989 (First for Women), so Arthur's Magazine was started first. Answer Arthur's Magazine

---

**Few-shot 6 Query:**
Were Pavel Urysohn and Leonid Levin known for the same type of work?
**Few-shot 6 Response:**
's think step by step. Pavel Urysohn is a mathematician. Leonid Levin is a mathematician and computer scientist. So Pavel Urysohn and Leonid Levin have the same type of work. Answer Yes

---

## D.2  RESPONSE LEVEL ANALYSIS

For each response selected by ReFeri, we compute the Eq. 11 for both best and worst response. The value in parentheses is the final selection score. The highest score among all candidates corresponds to the best response, and the lowest score represents the worst response. As shown in below examples, the selected response by ReFeri has more accurate reasoning. For instance, in the example of MATH-500, the best-scoring response anchor their reasoning in a coordinate system and follow clean logic. The worst response, on the other hand, ignore spatial cues, misapply subtraction, and over-complicate with lengthy and internally inconsistent steps. Similarly, in GPQA, the best response reasonably combine the results from the two analyses to arrive at the correct answer. The worst response, on the

other hand, appears to rely primarily on the idea that *"the heavy branching in (A) is consistent with the splitting observed"* without making a clear connection to the analyzed results. Thus, one response integrates the data, while the other reduces it to a vague notion of complexity. In HotpotQA, selected answer are consistent with HotpotQA's multi-hop requirements by making intermediate hops explicit and factually correct, while rejected answer provide unsupported single-hop claims.

---

Responses for MATH500:

---

**Test-Query**:
Twelve 1 by 1 squares form a rectangle, as shown. What is the total area of the shaded region?
[asy]
size(4cm);defaultpen(linewidth(0.75));

// Filled portions
fill((0, 4)–(0, 0)–(2, 0)–cycle, gray(0.75));
fill((0, 4)–(3, 4)–(3, 0)–cycle, gray(0.75));

// grid
int j;
for (j = 0; j < 4; ++j) draw((j, 0)–(j, 4));
for (j = 0; j < 5; ++j) draw((0, j)–(3, j));

//diagonals
draw((0, 4)–(3, 0)); draw((0, 4)–(2, 0));
[/asy]

---

---

Responses for MATH500:

---

**Worst Response (-1.736):**

To find the total area of the shaded region defined by the triangle at the top left and the rectangle on the right, we proceed as follows:

1. Identify the dimensions of the figure: The outline shows a rectangle formed by combining the squares. The rectangle consists of 3 squares in width (3 squares wide) and 4 squares in height (4 squares tall). Therefore, the rectangle has a total area of $3 \times 4 = 12$ square units.

2. Determine the area of the upper triangle: The shaded area in the upper left can be identified as a right triangle, located at the top left corner of the rectangle. The base of this triangle is the height of the rectangle (4 squares), and the height is the width of the rectangle (2 squares). The area of a triangle is computed using the formula:

$$\text{Area of triangle} = \frac{1}{2} \times \text{base} \times \text{height}$$

For this triangle:

$$\text{Area} = \frac{1}{2} \times 2 \times 4 = \frac{1}{2} \times 8 = 4 \text{ square units}$$

3. Determine the area of the other shaded region: The other shaded area is the rectangle that extends across the top down to the base of the triangle. The rectangle also covers one of the 1x1 squares fully, specifically, it will cover the upper two squares fully which are not part of the triangle. This gives us a square area of $3 \text{ (width)} \times 4 \text{ (height)} - \text{Area of triangle} = 12 - 4$. Thus, the shaded area for consideration is:

$$12 - 4 = 8 \text{ square units}$$

4. Calculate the total shaded area: Adding these two areas we've established gives: - Area of upper triangle: 4 - Area of square: 8

However, we must ensure we are not double counting any area. Since the rectangle is inclusive, we need only evaluate the total shaded area: The shaded area includes just the triangle area since it encompasses all squares minus the interfered structured area.
The area accounted for based on shapes left is 4 described above in figures and does not undermine new counted elements.

5. Final answer: Thus, the area of the shaded region is confirmed as:

$$\boxed{4}$$

---

Responses for MATH500:

**Best Response (-0.6992):**
To find the total area of the shaded region in the rectangle formed by twelve 1 by 1 squares, let's analyze the problem step by step.

1. Understanding the Shape: The given rectangle consists of three columns and four rows of squares, leading to a total of $3 \times 4 = 12$ unit squares.

2. Identifying the Shaded Areas: The shaded areas consist of two triangular regions: - The first triangular area is defined by points (0, 4), (0, 0), and (2, 0). - The second triangular area is defined by points (0, 4), (3, 4), and (3, 0).

3. Calculating the Area of the First Triangle: - Base = 2 (from (0, 0) to (2, 0)) - Height = 4 (from (0, 0) to (0, 4)) - Area of the triangle = $\frac{1}{2} \times$ Base $\times$ Height $= \frac{1}{2} \times 2 \times 4 = 4$.

4. Calculating the Area of the Second Triangle: - Base = 3 (from (3, 0) to (3, 4)) - Height = 4 (from (0, 4) to (0, 4)) - Area of the triangle = $\frac{1}{2} \times$ Base $\times$ Height $= \frac{1}{2} \times 3 \times 4 = 6$.

5. Adding the Areas: - Total shaded area = Area of first triangle + Area of second triangle = $4 + 6 = 10$.

Thus, the total area of the shaded region is $\boxed{10}$.

---

Responses for GPQA:

**Test-Query:**
Question: You have prepared an unknown compound. To identify the product, you have used the following characterisation techniques: FTIR and 1H NMR. The FTIR spectrum shows a very broad absorption peak at 3000 wavenumbers. A strong absorption peak is also observed at 1700 wavenumbers. Several peaks were observed in the 1H NMR spectrum, none of which are correspond to vinyl-hydrogens. One of the signals in the 1H NMR is a doublet of triplets of quartets whilst a different signal is a doublet of triplets of triplets. Identify the compound as either, CH3CH2C(H)(CH3)C(H)(CH3)COOH, CH3CH2C(H)(C2H5)C(H)(C2H5)COOH, CH3C(H)(CH3)C(H)(CH3)CH2COOH, or CH3C(H)(C2H5)C(H)(C2H5)CH2COOH

Responses for GPQA:

**Worst Response (-1.344):**
Let's analyze the information provided systematically.

1. **FTIR Analysis:**
- A broad absorption peak at 3000 wavenumbers indicates the presence of -OH or -NH groups, which are characteristic of carboxylic acids due to the hydrogen bonding which generates a broad peak.
- A strong absorption peak at 1700 wavenumbers suggests the presence of a carbonyl group (C=O), also associated with the carboxylic acid functional group.

2. **1H NMR Analysis:**
- The absence of vinyl hydrogen signals suggests that none of the compounds contain a double bond (i.e., C=C).
- The signal described as a "doublet of triplets of quartets" indicates complex splitting patterns, likely resulting from hydrogens on adjacent carbon atoms that are not equivalent due to the molecular structure.
- The presence of various integrations and patterns suggests alkyl groups and potentially multiple types of adjacent hydrogens.

3. **Evaluating the Choices:**
- Each choice represents a unique structure. We need to figure out which matches the spectral data:
- (A) CH3C(H)(CH3)C(H)(CH3)CH2COOH has the structure for 5 unique hydrogen environments and matches possible splitting patterns due to the complexity of the methyl and methylene groups.
- (B) CH3CH2C(H)(CH3)C(H)(CH3)COOH also has multiple potential splitting but is slightly less complex than (A).
- (C) CH3C(H)(C2H5)C(H)(C2H5)CH2COOH has an ethyl group that could create distinctive NMR peaks but still maintains hydrogen integration complexity.
- (D) CH3CH2C(H)(C2H5)C(H)(C2H5)COOH similarly maintains complexity, but having two ethyl groups may lead to a different integration pattern.

Given this information, we can further deduce that since the FTIR indicated a carboxylic acid and the NMR shows no vinyl protons, we can confirm the presence of -COOH in any of the compounds.

The heavy branching in (A) is consistent with the splitting observed (doublet of triplets of quartets) due to more complex interactions among various hydrogen types.

Considering the complexity and integration in the 1H NMR, we can conclude that the most likely compound based on the spectral data is (A) because of the correct splitting patterns associated with the branching substitutions.

The correct answer is (A) CH3C(H)(CH3)C(H)(CH3)CH2COOH.

Responses for GPQA:

**Best Response (-1.134):**
To identify the unknown compound based on the provided spectroscopic data, let's analyze the information step by step.

1. **FTIR Analysis**:
- A broad absorption peak at **3000 wavenumbers** indicates the presence of an -OH group, typical of carboxylic acids or alcohols.
- A strong absorption peak at **1700 wavenumbers** suggests a carbonyl (C=O) group, common in carboxylic acids and aldehydes.

2. **1H NMR Analysis**:
- The absence of vinyl hydrogens indicates there are no double bonds present in the compound.
- The **doublet of triplets of quartets** and a **doublet of triplets of triplets** indicate a relatively complex splitting pattern which typically arises from protons in different environments due to adjacent protons.

3. **Analysis of the compounds**:
- The compounds listed can be broken down as follows:
- (A) CH3C(H)(CH3)C(H)(CH3)CH2COOH: A branched structure likely leading to complex splitting.
- (B) CH3CH2C(H)(CH3)C(H)(CH3)COOH: A linear structure with groups that may contribute to the complexity in splitting.
- (C) CH3C(H)(C2H5)C(H)(C2H5)CH2COOH: Also branched, with ethyl groups that could lead to intricate splitting.
- (D) CH3CH2C(H)(C2H5)C(H)(C2H5)COOH: A linear structure including ethyl groups, likely resulting in simpler splitting.

4. **Choosing the Compound**:
- The broad absorption at 3000 cm-1 and significant peaks at 1700 cm-1 underscores that we are likely dealing with a carboxylic acid, given it shows a carbonyl stretch and alcohol absorbance.
- The complexity of the NMR signals suggests a branched structure as those are usually more complex due to the adjacent hydrogen interactions.

5. **Final Decision**:
- **Compound (C)**, containing ethyl groups and overall complex splitting, aligns with the described spectroscopic data. The branching and additional substituents in compound (C) can contribute to complex resonance, further supported by the FTIR properties.

Thus, based on the analysis of FTIR and NMR data, the compound can be most accurately identified as:

The correct answer is (C).

---

Responses for HotpotQA:

**Test-Query**:
H-63 is a highway that runs near the county seat of what Michigan county?.

Responses for HotpotQA:

**Worst Response (-1.324):**
Let's think step by step. The H-63 highway runs in Michigan near the county seat of Baraga County, which is L'Anse. So, H-63 is associated with Baraga County.

Answer Baraga County

**Best Response (-1.033):**
Let's think step by step. H-63 is a highway located in Michigan. To determine the county seat it runs near, we can look at its route. H-63 runs primarily through Chippewa County in the Upper Peninsula of Michigan. The county seat of Chippewa County is Sault Ste. Marie. Therefore, H-63 is a highway that runs near the county seat of Chippewa County.

Answer Chippewa County

**Listing 1** Few-shot CoT prompt on MATH500

```
f'''
{system}
Please reason step by step, and put your final answer within \boxed{{}}.
-----------------------------------------------
{user}
Kevin Kangaroo begins hopping on a number line at 0. He wants to get to
↪   1, but he can hop only $\frac{1}{3}$ of the distance. Each hop tires
↪   him out so that he continues to hop $\frac{1}{3}$ of the remaining
↪   distance. How far has he hopped after five hops? Express your answer
↪   as a common fraction.

Let's think step by step
Kevin hops $1/3$ of the remaining distance with every hop.
His first hop takes $1/3$ closer.
For his second hop, he has $2/3$ left to travel, so he hops forward
↪   $(2/3)(1/3)$.
For his third hop, he has $(2/3)^2$ left to travel, so he hops forward
↪   $(2/3)^2(1/3)$.
In general, Kevin hops forward $(2/3)^{k-1}(1/3)$ on his $k$th hop.
We want to find how far he has hopped after five hops.
This is a finite geometric series with first term $1/3$, common ratio
↪   $2/3$, and five terms.
Thus, Kevin has hopped
↪   $\frac{\frac{1}{3}\left(1-\left(\frac{2}{3}\right)^5\right)}
{1-\frac{2}{3}} = \boxed{\frac{211}{243}}$.
The answer is \frac{211}{243}}

...

Convert the point $(0,3)$ in rectangular coordinates to polar
↪   coordinates.  Enter your answer in the form $(r,\theta),$ where $r >
↪   0$ and $0 \le \theta < 2 \pi.$
'''
```

**Listing 2** Zero-shot CoT prompt on MATH500

```
f'''
{system}
Please reason step by step, and put your final answer within \boxed{{}}.
--------------------------------------------------
{user}
Convert the point $(0,3)$ in rectangular coordinates to polar
↪  coordinates.  Enter your answer in the form $(r,\theta),$ where $r >
↪  0$ and $0 \le \theta < 2 \pi.$
'''
```

**Listing 3** Prompt for USC

```
f'''
I have generated the following responses to the question: Convert the
↪  point $(0,3)$ in rectangular coordinates to polar coordinates.
↪  Enter your answer in the form $(r,\theta),$ where $r > 0$ and $0 \le
↪  \theta < 2 \pi.$

Response 0: {response0}

...

Response 4: {response4}

Evaluate these responses.
Select the most consistent response based on majority consensus.
Start your answer with "The most consistent response is Response X"
↪  (without quotes).
'''
```

**Listing 4** Prompt for LEAP mistakes

```
f'''
{system}
Please reason step by step, and put your final answer within \boxed{{}}.
--------------------------------------------------
{user}
Kevin Kangaroo begins hopping on a number line at 0. He wants to get to
↪  1, but he can hop only $\frac{1}{3}$ of the distance. Each hop tires
↪  him out so that he continues to hop $\frac{1}{3}$ of the remaining
↪  distance. How far has he hopped after five hops? Express your answer
↪  as a common fraction.
'''
```

**Listing 5** Prompt for LEAP low-level principles

```
f'''
Question: {question}
Generated Reasoning: {response}

Generated Answer: {generated_answer}

Correct Reasoning: {correct_reasoning}

Correct Answer: {correct_answer}

Instruction: Conduct a thorough analysis of the generated answer in
↪   comparison to the correct answer. Also observe how the generated
↪   reasoning differs from the correct reasoning. Identify any
↪   discrepancies, misunderstandings, or errors. Provide clear insights,
↪   principles, or guidelines that can be derived from this analysis to
↪   improve future responses. We are not focused on this one data point,
↪   but rather on the general principle.

Reasoning: <discuss why the generated answer is wrong>
Insights: <what principle should be looked at carefully to improve the
↪   performance in the future>

'''
```

**Listing 6** Prompt for LEAP high-level principles

```
f'''
Low-level principles:
{low_level_principles}

Create a list of *unique* and insightful principles to improve future
↪   responses based on the analysis above.
Focus on capturing the essence of the feedback while eliminating
↪   redundancies.
Ensure that each point is clear, concise, and directly derived from the
↪   introspection results.
Create a numbered list of principles. Leave specific details in place.
Limit to at most 8 principles.

List of Principles:
'''
```

**Listing 7** Prompt for LEAP generations

```
f'''
{system}
Please reason step by step, and put your final answer within \boxed{{}}.
-------------------------------------------------
{user}
Please carefully note the following principles:

Principles: 1. **Meticulous Verification**: Always verify each step in
↪    algebraic processes to prevent errors that can lead to incorrect
↪    conclusions.

...

8. **Continuous Learning and Adaptation**: Stay open to learning from
↪    mistakes and adapting methods to improve future problem-solving
↪    approaches.

Kevin Kangaroo begins hopping on a number line at 0. He wants to get to
↪    1, but he can hop only $\frac{1}{3}$ of the distance. Each hop tires
↪    him out so that he continues to hop $\frac{1}{3}$ of the remaining
↪    distance. How far has he hopped after five hops? Express your answer
↪    as a common fraction.

Let's think step by step
Kevin hops $1/3$ of the remaining distance with every hop.
His first hop takes $1/3$ closer.
...

Convert the point $(0,3)$ in rectangular coordinates to polar
↪    coordinates.  Enter your answer in the form $(r,\theta),$ where $r >
↪    0$ and $0 \le \theta < 2 \pi.$
'''
```

**Listing 8** Prompt for USC-w/ Fewshot

```
f'''
Kevin Kangaroo begins hopping on a number line at 0. He wants to get to
↪  1, but he can hop only $\frac{1}{3}$ of the distance. Each hop tires
↪  him out so that he continues to hop $\frac{1}{3}$ of the remaining
↪  distance. How far has he hopped after five hops? Express your answer
↪  as a common fraction.

Let's think step by step
Kevin hops $1/3$ of the remaining distance with every hop.
His first hop takes $1/3$ closer.
...

I have generated the following responses to the question: Convert the
↪  point $(0,3)$ in rectangular coordinates to polar coordinates.
↪  Enter your answer in the form $(r,\theta),$ where $r > 0$ and $0 \le
↪  \theta < 2 \pi.$

Response 0: {response0}

...

Response 4: {response4}

Evaluate these responses.
Select the most consistent response based on majority consensus.
Start your answer with "The most consistent response is Response X"
↪  (without quotes).
'''
```

**Listing 9** Prompt for LLM-as-Judge

```
f'''
{system}
Your job is selecting the most accurate response among multiple
↪  candidates. You will receive a question and several candidate
↪  answers labeled candidate1, candidate2, etc. Please summarize the
↪  debate very briefly and then conclude which single candidate is the
↪  most plausible. Output exactly in this format:
Summary: <brief summary>
Conclusion: candidate<number>
Remember to choose only one candidate as the final answer.
-------------------------------------------------
{user}
Please reason step by step, and put your final answer within \boxed{{}}.

The below examples are well-constructed gold question and answer pairs
↪  for the same task.

Kevin Kangaroo begins hopping on a number line at 0. He wants to get to
↪  1, but he can hop only $\frac{1}{3}$ of the distance. Each hop tires
↪  him out so that he continues to hop $\frac{1}{3}$ of the remaining
↪  distance. How far has he hopped after five hops? Express your answer
↪  as a common fraction.

Let's think step by step
Kevin hops $1/3$ of the remaining distance with every hop.
His first hop takes $1/3$ closer.
...

Now, let's select the most proper answer for the given question
Question: Convert the point $(0,3)$ in rectangular coordinates to polar
↪  coordinates.  Enter your answer in the form $(r,\theta),$ where $r >
↪  0$ and $0 \le \theta < 2 \pi.$
candidate1: {response 0}
...
candidate5: {response 4}
'''
```

**Listing 10** Prompt for generate weak few-shot

```
f'''
 "You will receive a QUESTION and its original ANSWER.\n"
"Rewrite ONLY the ANSWER; do NOT alter the QUESTION.\n"
"Treat the original as a 10/10 reference. Produce a deliberately
↪  degraded explanation (target quality 1/10):\n"
"- Keep the final answer tokens EXACT (e.g., '\\boxed{...}' or 'The
↪  correct answer is (X)').\n"
"- Keep the original CoT style label if present (e.g., 'Let's think step
↪  by step:' / 'Reasoning:').\n"
"- Make reasoning weak: shallow, vague, incomplete; omit steps, avoid
↪  precise formulas/numbers.\n"
"- Prefer generic phrases over concrete derivations. Lower clarity and
↪  rigor compared to the original.\n"
"OUTPUT FORMAT: Return EXACTLY ONE JSON object and NOTHING ELSE:\n"
'{"answer":"<rewritten weaker answer>"}'
'''
```

**Listing 11** Prompt for judging weak few-shot

```
f'''
 "You are judging FEW-SHOT QUALITY only.\n"
"Compare TWO blocks side-by-side. Assume BLOCK A and BLOCK B are
↪  candidate few-shot demonstrations.\n\n"
"Ignore question quality entirely -- the question is context only.\n\n"

"What "answer quality" means here:\n"
"- clarity, structure, and coherence of the reasoning.\n"
"- specific steps, concrete numbers/equations when relevant, and
↪  justified transitions.\n"
"- a single, clearly marked final answer token format (e.g.,
↪  \"\\boxed{...}\" or \"The correct answer is (X)\") if present;\n"

"Instructions:\n"
"- Assign an integer score 1-10 to EACH block (higher = better few-shot
↪  quality).\n"
"- The evaluation should be comparative: scores must reflect their
↪  relative quality.\n"
"- Provide brief notes explaining each score.\n\n"
"OUTPUT FORMAT:\n"
"Return exactly ONE JSON object with this schema (and nothing else):\n"
"{"
"\"A\":{\"score\":int,\"notes\":string},"
"\"B\":{\"score\":int,\"notes\":string},"
"\"comparative_notes\":string"
'''
```

**Listing 12** Prompt for plan-and-sovle on MATH500

```
f'''
"Let's first understand the problem, extract relevant variables and
↪  their corresponding numerals, and make a complete plan. Then, let's
↪  carry out the plan, calculate intermediate variables (pay attention
↪  to correct numerical calculation and commonsense), solve the problem
↪  step by step, and put your final answer within \\boxed{{}}.\n"
'''
```

**Listing 13** Prompt for role-playing on MATH500

```
f'''
"From now on, you are an excellent math teacher and always teach your
↪  students math problems correctly. And I am one of your students. Put
↪  your final answer within \\boxed{{}}.\n"
'''
```

