# OpenReview forum: "Training-free LLM Verification via Recycling Few-shot Examples"
_ICLR.cc/2026/Conference — Submitted to ICLR 2026_

### Official Review · Reviewer_UCFk · 2025-10-23

**Soundness:** 1
**Presentation:** 2
**Contribution:** 2
**Rating:** 2
**Confidence:** 4

**Summary:**

The paper presents ReFeri, a training-free framework for selecting the best output from multiple LLM generations by repurposing few-shot examples for verification, using a Bayes' rule-derived scoring function. While the idea of leveraging Bayes’ rule to combine forward and backward likelihoods is conceptually elegant, the claimed novelty is overstated.

**Strengths:**

1. The core idea of "recycling" few-shot examples for verification, not just generation, is  a novel perspective.

2. The experimental design is a major strength. The use of seven diverse benchmarks and three LLMs provides compelling evidence for the method's generalizability.

**Weaknesses:**

1. The proposed method is a re-interpretation of likelihood-based reranking and LLM-as-judge techniques, both of which already use internal model probabilities or few-shot conditioning. The paper fails to clearly delineate how ReFeri fundamentally differs from prior “Best-of-N” and likelihood-based methods (e.g., CoT-WP, PRM, ORM, or self-consistency with verification). The novelty is slightly diminished by the existence of methods like CoT-WP, which already use the forward score. The primary innovation then becomes the introduction of the backward score.

2. The connection to Bayes' rule feels somewhat post-hoc and tenuous. Eq. (5) is a mathematical tautology, but the paper does not sufficiently justify why decomposing and estimating these two terms separately is more effective than directly modeling P(y_k), especially given the admitted "discrepancies" due to model limitations. The theoretical motivation lacks a solid, empirical or theoretical foundation for why it works better in practice.

3. The reported 1–2% improvement over the strongest baselines is statistically weak, especially given that the comparison baselines are limited to prompt-based methods. No evidence is provided that ReFeri generalizes beyond the selected reasoning datasets. The improvements reported (Table 1) lack standard deviation or confidence intervals. Given the stochasticity of LLM sampling, it’s unclear whether differences of 1–2% are meaningful.

4.The intuition behind the backward score is explained at a high level ("how well the candidate explains the few-shot examples"), but its mechanistic interpretation is vague. Does a high backward score imply the candidate response is a good "archetype" or "rule" that generalizes to the few-shot examples?

5. The "lightweight approximation" is a major compromise. The decision to use only the single most relevant example for the backward score (Eq. 9-10) for computational reasons is understandable but undermines the core Bayesian formulation. The method is no longer a true application of Eq. (5) but a heuristic approximation. Why this approximation does not collapse the entire theoretical framework?

6. Although the method is claimed to be task-agnostic, all experiments are conducted on structured QA tasks. Open-ended tasks (e.g., summarization, dialogue) are not explored, where “verification” is more challenging.

7. The estimation model is ambiguity. The method relies heavily on a “likelihood estimation model,” yet it is never discussed how this model differs from the generation LLM, how temperature affects estimation, or how token-level log-probabilities are normalized across different lengths.

8. Although the method is claimed to be “training-free,” it involves multiple LLM calls per candidate and per example, which can be expensive. No clear runtime or scaling comparison with existing test-time reranking methods is given.

**Questions:**

1. Does a high backward score imply the candidate response is a good "archetype" or "rule" that generalizes to the few-shot examples?

2. Why the lightweight approximation does not collapse the entire theoretical framework?

---

> ### Author Response · Authors · 2025-11-22
> **Response to Reviewer UCFk [1/5]**
>
> Dear Reviewer UCFk,
>
> We appreciate your review with thoughtful comments. We have carefully considered each of your questions and provide detailed responses below.
>
> ---
> **[W1 & W2] Novelty and Theoretical motivation**
>
> ReFeri is **NOT** a re-interpretation of existing likelihood-based reranking or LLM-as-judge techniques. Our method is driven by a distinct research question: **how can few-shot examples be leveraged not only for generation, but also for verification?** Few-shot examples remain one of the most practical ways to inject human prior knowledge into LLMs, especially for tasks or domains where the model’s internal knowledge is insufficient. However, existing approaches, including likelihood-based rerankers (e.g., CoT-WP), LLM-as-judge systems, and few-shot prompting, use these examples only as additional input context during forward LLM. Consequently, these methods rely entirely on the model’s intrinsic in-context learning ability and provide **no explicit mechanism** to verify or evaluate candidate responses based on the same few-shot demonstrations.
>
> Motivated by this gap, we introduce a **Bayes-inspired decomposition of the selection objective** (Eq. 5), yielding two complementary terms: a forward confidence score and a backward consistency score, both conditioned on few-shot examples. This decomposition is not a heuristic combination of probabilities but a principled transformation grounded in Bayes’ rule, which clarifies the roles of the two terms.
>
> Intuitively, the two terms play fundamentally different roles and complement each other:
> - The forward score measures how likely the candidate is under the distribution induced by the few-shot context.
> - The backward score, which is novel to our work, measures how well the candidate aligns with the underlying rules, structures, or patterns encoded in the few-shot demonstrations.
>
> While forward likelihood has been explored in previous work (e.g., CoT-WP can be viewed as a special case focusing only on the answer portion), the backward score is entirely novel and follows directly from our Bayes-based formulation. This term enables a new capability: explicit verification using the few-shot examples, in contrast to prior training-free reranking methods that only rely on internal model probabilities.
>
> We also emphasize that the effectiveness of this backward term is empirically validated. Across all evaluated settings, ReFeri consistently outperforms forward-only baselines such as CoT-WP and Self-Certainty, demonstrating that the Bayes decomposition is not merely theoretical but yields practical improvements.
>
> Therefore, the contribution and novelty of ReFeri are not diminished by existing methods. Whereas previous work focuses on how to extract or manipulate internal model probabilities, our work introduces a new, theoretically grounded framework that reuses few-shot examples in a dual role, generation and verification, which has not been explored in prior research.
>
> ---
> **[W3] Given the stochasticity of LLM sampling**
>
> We would like to clarify several important points and provide additional evidence.
>
> **#1. Comparison baselines are limited to prompt-based methods**
>
> We would like to clarify that our baselines are NOT limited to prompt-based methods. Alongside prompt-based reranking (e.g., USC), we also evaluate likelihood-based approaches such as CoT-WP, and we additionally include Self-Certainty [1] in the revised draft (Table 11) following other reviewer’s suggestions. These baselines cover the major categories of training-free test-time selection methods.
>
> Our baseline choices were guided by our scope: ReFeri aims to be a training-free reranking method, so we compare against methods that do not require training reward models or task-specific supervision. We believe this provides a fair and meaningful comparison within the intended setting. That said, if the reviewer is aware of additional training-free methods that we may have missed, we would be happy to include them during the discussion period.
>
> **#2. No evidence is provided that ReFeri generalizes beyond the selected reasoning datasets**
>
> We refer the reviewer to our response in [W6], where we extend our evaluation to the open-ended LLM personalization tasks in LaMP-4 (personalized news headline generation) and LaMP-5 (personalized scholarly title generation). These tasks are not structured QA, and yet ReFeri consistently improves ROUGE-based metrics, demonstrating its applicability beyond reasoning datasets.

---

> ### Author Response · Authors · 2025-11-22
> **Response to Reviewer UCFk [2/5]**
>
> **#3. Lack of standard deviation or confidence intervals (continued for [W3])**
>
> The baseline methods (e.g., USC, CoT-WP), as well as ReFeri, all operate deterministically on the same fixed set of candidate responses. Therefore, once the candidate set is fixed, there is no source of randomness in the scoring process, and the reported results inherently have no variance. This is why standard deviations were not originally included in the main table.
> In addition, we remark that the variability arising from the stochastic generation process has already been evaluated through multiple independent runs in Table 9 of the original draft. These results measure variability across independently sampled candidate sets and compare performance against CoT-WP.
>
> Nevertheless, to directly address the reviewer’s concern, we repeated all experiments in Table 1 for two additional runs, yielding three independent trials in total. The updated Table 11 in the revision now includes the complete results along with standard deviations. Across all cases, ReFeri consistently outperforms the baselines (e.g., CoT-WP, USC). For example, with GPT-4o-mini, ReFeri achieves an average accuracy of 63.9 ± 0.5, outperforming USC (62.3 ± 0.5). Similarly, for GPT-4o, ReFeri reaches 71.5 ± 0.4, exceeding the newly added Self-Certainty baseline (69.7 ± 0.4).
>
> Overall, these results confirm that the improvements achieved by ReFeri are statistically reliable.
>
> ---
>
> **[W4, Q1] Does a high backward score imply the candidate response is a good "archetype" or "rule" that generalizes to the few-shot examples?**
>
> Thank you for the insightful comment regarding the interpretation of the backward score. First, we note the qualitative token-level analysis presented in Appendix D.1. In that section, we visualize which tokens in the few-shot examples contribute most to the backward score. These analyses show that the backward term selectively attends to task-relevant components; for example, numerical symbols or answer-format tokens in MATH500 and GPQA.
>
> Next, to further address the reviewer’s question, we conducted an additional experiment designed to test whether a candidate with a high backward score indeed captures a task-level “rule” or “archetype.” Specifically, we first took the generated outputs from LLaMA, ranked them by their backward score, and used each output as a one-shot demonstration to solve the original few-shot queries. For a more intuitive understanding, we used the no-replace backward score, which is a simplified variant where each few-shot example $\mathbf{x_i}$ is evaluated in a one-shot manner using test query $\tilde q$ and the candidate response $r_k$ as condition:
>
> $S^\prime_{\texttt{Back}}(r_k) := \log P(a_{i^\dagger} \vert q_{i^\dagger}, \tilde{q}, r_k) - \log P(a_{i^\dagger} \vert q_{i^\dagger})$
>
> By utilizing the no-replace term, we isolate the specific impact of the candidate response $r_k$ on measuring a single few-shot example $x_i$, thereby eliminating the confounding factor of other few-shot demonstrations.
>
> \begin{array}{l|ccccc} \hline
> \text{Task}
> & 1 & 2 & 3 & 4 & 5 \newline \hline
> \text{MATH500}
> & 87.0 & 85.4 & 85.2 & 84.2 & 84.5 \newline
> \text{MuSR-ta}
> & 37.2 & 37.6 & 33.7 & 30.3 & 29.9 \newline \hline
> \end{array}
>
> As shown in the table above, we observe a clear correlation: responses with higher backward scores more effectively guide the model to answer the original few-shot examples. This indicates that the backward term is not merely a heuristic signal, but it reflects whether the candidate serves as a strong archetype for the underlying reasoning pattern present in the demonstrations.
>
> In other words, a high backward score suggests that the candidate not only aligns with the task distribution but also encapsulates the reasoning structure that generalizes across the few-shot examples. We will add this discussion to the revision for clarity in Appendix B.9.
>
> ---
>
> **[W5, Q2] The “lightweight approximation” is a major compromise. Why does it not collapse the theoretical framework?**
>
> We acknowledge that the lightweight approximation (Eq. 9–10) does not strictly preserve the full Bayesian form in Eq. (5), since it uses only the single most relevant example rather than the full set of demonstrations. However, our intention is not to claim that a single example is universally sufficient or that multiple demonstrations are unimportant. Rather, the lightweight variant is designed to provide a **computationally efficient approximation** while **retaining the conceptual structure of ReFeri**. The full formulation remains available and theoretically valid; the lightweight version is simply an efficient surrogate when cost is a practical concern.

---

> ### Author Response · Authors · 2025-11-22
> **Response to Reviewer UCFk [3/5]**
>
> **[W5, Q2] The “lightweight approximation” is a major compromise. Why does it not collapse the theoretical framework? (con't)**
>
> Importantly, prior work in in-context learning has repeatedly shown that the contribution of few-shot examples is highly uneven, and that the most relevant example often carries a disproportionately large fraction of the useful signal [2,3,4]. From this perspective, selecting the most relevant demonstration does not replace the conceptual role of the backward term; it simply offers a tractable proxy that captures most of the signal. Our empirical results support this view: as shown in Table 18 (for convenience, we reproduce this table below), the performance gap between the full backward computation and the lightweight approximation is negligible across benchmarks, even though the lightweight version substantially reduces computation.
>
> \begin{array}{l|ccccc c c|c} \hline
> \text{Methods}
> & \text{MuSR-ta}
> & \text{MuSR-op}
> & \text{GPQA}
> & \text{MATH500}
> & \text{DROP (EM/F1)}
> & \text{HotpotQA (EM/F1)}
> & \text{MMLU-Pro}
> & \text{Avg.} \newline \hline
> \text{No replace (full)}
> & \mathbf{82.8} & 60.2 & 42.4 & \mathbf{78.0}
> & 78.4/84.2 & \mathbf{36.2/48.0} & \mathbf{65.0} & 63.3 \newline
> \text{No replace}
> & 82.4 & 60.2 & \mathbf{42.9} & 77.6
> & 78.4/84.1 & 35.8/47.6 & 64.7 & 63.1 \newline \hline
> \text{ReFeri (Full)}
> & \mathbf{82.8} & \mathbf{61.3} & 42.4 & 77.8
> & \mathbf{79.6/85.3} & 35.8/47.9 & \mathbf{65.0} & \mathbf{63.5} \newline
> \text{ReFeri}
> & \mathbf{82.8} & \mathbf{61.3} & 41.9 & 77.8
> & 79.2/84.9 & \mathbf{36.2/48.0} & 64.9 & 63.4 \newline \hline
> \end{array}
>
>
> Moreover, we highlight that our method is applicable without losing the theoretical validity while keeping the moderate computation, thanks to its weak reliance on estimation model size. To demonstrate this approach, we conducted additional experiment with ReFeri (Full) in addition to the experiments in Table 17, and the results are presented in below:
>
> \begin{array}{c|c|ccc} \hline
> \text{size} & \text{Methods} & \text{MATH500}  & \text{GPQA} & \text{MuSR-ta} \newline
> & & \text{(Acc(\\%) / Time(s))} &  \text{(Acc(\\%) / Time(s))} &  \text{(Acc(\\%) / Time(s))}\newline  \hline
> \text{1B} & \text{USC} & {75.0 / 0.6} & {44.9 / 0.1} & {75.6 / 0.7} \newline
> & \text{CoT-WP} & {76.0 / 1.5} & {43.4 / 2.0} & {77.6 / 5.0} \newline
> & \text{ReFeri (Full)} & {78.0 / 9.6} & {44.9 / 12.6} & {83.2 / 21.3} \newline
> & \text{ReFeri} & {78.0 / 3.0} & {43.9 / 4.0} & {83.2 / 8.0} \newline  \hline
> \text{8B} & \text{USC} & {77.8 / 3.7} & {46.0 / 3.7} & {74.4 / 3.9} \newline
> & \text{CoT-WP} & {77.8 / 8.3} & {42.4 / 11.0} & {78.8 / 25.6} \newline
> & \text{Referi} & {77.8 / 16.6} & {41.9 / 22.1} & {82.8 / 41.8} \newline \hline
> \end{array}
>
> Even with full backward computation, ReFeri Full (1B) achieves stronger accuracy than CoT-WP using an 8B estimator despite requiring similar or lower computation. For example, on MuSR-ta, ReFeri Full (1B) achieves 83.2% accuracy with a cost of 21.3s per query, whereas CoT-WP (8B) achieves 78.8% accuracy with 25.6s. In contrast, CoT-WP experiences substantial degradation when reducing the estimator from 8B to 1B, while ReFeri maintains stable performance across estimator scales.
>
> These findings demonstrate two key points:
>
> 1. **The full Bayesian structure remains unchanged and practically usable**: the lightweight version is optional, not foundational.
> 2. **When used, the lightweight approximation preserves the essential function of the backward term** and does not collapse the theoretical framework, as supported by both the observations in prior works and empirical evidence.
>
> We appreciate the reviewer’s insightful comment. We have added this discussion, along with the new results, to Appendix B.8 of the revised draft and will incorporate it into the camera-ready version.

---

> ### Author Response · Authors · 2025-11-22
> **Response to Reviewer UCFk [4/5]**
>
> **[W6] Open-ended tasks (e.g., summarization, dialogue) are not explored, where “verification” is more challenging**
>
> We appreciate the reviewer’s valuable suggestion. To address this concern, we extended our evaluation to open-ended generation tasks to demonstrate that ReFeri’s verification mechanism generalizes beyond structured QA. Specifically, we applied ReFeri to an LLM personalization setting, where the goal is to steer an LLM’s responses to better match an individual user’s writing preferences, an inherently open-ended generation scenario.
>
> A representative baseline for personalization is few-shot retrieval-augmented generation (RAG), which retrieves a user’s previous writing examples. This makes it a natural setting to apply ReFeri. We evaluate on two tasks from the LaMP benchmark [5], LaMP-4 (personalized news headline generation) and LaMP-5 (personalized scholarly title generation), using GPT-4o-mini as the generation model. We sample K=5 candidate responses with temperature 1.0 and use N=3 examples for both Few-shot RAG and ReFeri. The results are shown below:
>
> \begin{array}{l|cc|cc} \hline
> \text{Methods}
> &  {\text{LaMP-4}}  &  &{\text{LaMP-5}} \newline
> & \text{Rouge-1} & \text{Rouge-L} & \text{Rouge-1} & \text{Rouge-L} \newline \hline
> \text{Vanilla}        & 0.120 & 0.106 & 0.421 & 0.332 \newline
> \text{Few-shot RAG}   & 0.138 & 0.123 & 0.451 & 0.366 \newline \hline
> \text{Ours}           & \mathbf{0.143} & \mathbf{0.128}  & \mathbf{0.470} & \mathbf{0.394} \newline \hline
> \end{array}
>
> Across both tasks, ReFeri consistently improves ROUGE-1 and ROUGE-L scores compared to Vanilla and Few-shot RAG baselines (e.g., improving LaMP-5 ROUGE-1 from 0.451 to 0.470). These results confirm that ReFeri is applicable to open-ended generation tasks where exact answer matching is impossible, supporting our claim that the method is task-agnostic. We have added this result and discussion in Appendix B.3.
>
> ---
>
> **[W7] The estimation model is ambiguity**
>
> We appreciate the reviewer for highlighting this important point. We address each question below.
>
> **#1. How the estimation model differs from the generation LLM**
>
> As noted in line 141, footnote 2, and lines 262–263 of the original draft, the estimation model is a separate pre-trained LLM that is fully decoupled from the generation LLM. In our main experiments, the generative models are GPT-4o, GPT-4o-mini, and LLaMA-3.1-8B-Instruct, whereas scoring is performed by a separate model, LLaMA-3.1-8B-Instruct.
> This modular design is intentional, as it allows ReFeri to operate with closed-source/API-based generators that do not provide token probabilities, and it enables the use of a much smaller estimator to reduce computational cost without sacrificing performance.
>
> **#2. How temperature affects estimation**
>
> During scoring with the estimation model, we use raw logits from the estimation model without applying additional temperature scaling.
> However, to address the reviewer's concern, we conducted experiments in which we scaled logits with scaling hyperparameter $T \in \{0.5, 1.0, 1.5, 2.0\}$. As shown in the table below, ReFeri remains highly stable across all settings, indicating that performance does not depend on the precise calibration of the estimation model. We have added this result and discussion in Appendix B.10.
>
> \begin{array}{c|c|ccccc}
> \hline
> \text{Models} & \text{Temp} & \text{MuSR-ta} & \text{MuSR-op} & \text{GPQA} & \text{MATH500} & \text{DROP (EM/F1)} & \text{HotpotQA (EM/F1)}  & \text{Avg.} \newline \hline
> \text{LLaMA-3.1-8B} & 0.5 & 79.6 & 57  & 33.3 & 50.6 & 69.8 /75.9  & 24.6 /33.0  & 52.5 &  \newline
> & 1   & 79.6 & 57.8 & 35.4 & 51.2 & 69.4 / 75.7 & 25.0 / 33.2 & 53.1 & \newline
> & 1.5 & 78.8 & 57   & 34.8 & 52   & 70.2 /76.1  & 25.2 /33.2  & 53.0 &  \newline
> & 2   & 78   & 56.2 & 34.3 & 51.8 & 71.0 /77.1  & 24.6 /32.7  & 52.7 &  \newline\hline
> \text{GPT-4o-mini} & 0.5 & 82   & 59.4 & 43.9 & 78   & 78.0 /83.8  & 35.8 /47.1  & 62.9 &  \newline
> & 1   & 82.8 & 61.3 & 41.9 & 77.8 & 79.2 / 84.9 & 36.2 / 48.0 & 63.2 &  \newline
> & 1.5 & 83.6 & 60.2 & 43.4 & 78   & 78.0 /84.7  & 36.4 /48.3  & 63.3 &  \newline
> & 2   & 83.2 & 59   & 41.4 & 78.4 & 77.4 /84.1  & 35.2 /47.6  & 62.4 &  \newline\hline
> \text{GPT-4o} & 0.5 & 90.4 & 71.1 & 52   & 77.4 & 84.0 /91.3  & 47.6 /60.7  & 70.4 &  \newline
> & 1   & 90.4 & 71.9 & 51.5 & 77.8 & 83.6 / 91.1 & 47.0 / 60.7 & 70.4 &  \newline
> & 1.5 & 91.2 & 71.1 & 50.5 & 77.6 & 83.4 /91.1  & 45.0 /59.2  & 69.8 &  \newline
> & 2   & 90.4 & 71.9 & 51   & 77.6 & 81.8 /89.9  & 44.8 /59.3  & 69.6 &  \newline\hline
> \end{array}
>
> **#3. How token-level log-probabilities are normalized across different lengths.**
>
> To avoid length bias when computing both the forward and backward scores, we use the average token-level log probability rather than the sum. We have updated Section 3 (lines 177–180) in the revision to explicitly describe this normalization step.

---

> ### Author Response · Authors · 2025-11-22
> **Response to Reviewer UCFk [5/5]**
>
> **[W8] No clear runtime or scaling comparison with existing test-time reranking methods is given.**
>
> We would like to clarify that detailed cost and runtime comparisons were already provided in Appendix B.6 (Table 15) of the original draft. For convenience, we reproduce Table 15 below:
>
> \begin{array}{c|c|ccc} \hline
> \text{size} & \text{Methods} & \text{MATH500}  & \text{GPQA} & \text{MuSR-ta} \newline
> & & \text{(Acc(\\%) / Time(s))} &  \text{(Acc(\\%) / Time(s))} &  \text{(Acc(\\%) / Time(s))}\newline  \hline
> \text{1B} & \text{USC} & {75.0 / 0.6} & {44.9 / 0.1} & {75.6 / 0.7} \newline
> & \text{CoT-WP} & {76.0 / 1.5} & {43.4 / 2.0} & {77.6 / 5.0} \newline
> & \text{ReFeri} & {78.0 / 3.0} & {43.9 / 4.0} & {83.2 / 8.0} \newline  \hline
> \text{8B} & \text{USC} & {77.8 / 3.7} & {46.0 / 3.7} & {74.4 / 3.9} \newline
> & \text{CoT-WP} & {77.8 / 8.3} & {42.4 / 11.0} & {78.8 / 25.6} \newline
> & \text{Referi} & {77.8 / 16.6} & {41.9 / 22.1} & {82.8 / 41.8} \newline \hline
> \end{array}
>
> As discussed in [W5], ReFeri with a small estimation model (1B) is highly competitive against baselines that rely on much larger estimation models (8B), and hence becomes even more computationally efficient while maintaining strong performance. For instance, ReFeri with a 1B estimator achieves 2.0% higher accuracy than CoT-WP with an 8B estimator, while being 67% faster in inference time on average. In contrast, CoT-WP exhibits substantial performance degradation when the estimator is reduced to 1B, indicating that it cannot easily lower computational cost by using a smaller estimator.
>
> More importantly, we clarify that multiple LLM calls per candidate are inherent to all test-time verification methods, not specific to ReFeri. Furthermore, while the original backward term requires one LLM call per few-shot example, we explicitly mitigate this cost with the computationally efficient approximation in Eq. (10). With this approximation, ReFeri requires only a single additional call to the estimation model to compute the backward score, regardless of the number of few-shot examples. As shown in Table 16, this estimation model can be extremely lightweight (e.g., 1B), making the overhead negligible compared to baselines that rely on large generator-sized verifiers.
>
> Overall, these results indicate that ReFeri provides a more favorable cost–performance tradeoff than existing methods, while maintaining robust accuracy even under low-cost inference settings.
>
> ---
>
> [1] Kang et al, Scalable Best-of-N Selection for Large Language Models via Self-Certainty, 2025
> [2] Liu et al., What Makes Good In-Context Examples for GPT-3? arXiv:2101.06804
> [3] Li et al., Compressing Context to Enhance Inference Efficiency of Large Language Models EMNLP 2023
> [4] Wang et al., Learning to Retrieve In-Context Examples for Large Language Models EACL 2024
> [5] Salemi et al., Lamp: When large language models meet personalization. ACL2024
>
> ---
>
> If you have any questions/concerns, please let us know.
>
> Thank you very much,
> Authors

---

### Official Review · Reviewer_htr2 · 2025-10-31

**Soundness:** 2
**Presentation:** 2
**Contribution:** 2
**Rating:** 4
**Confidence:** 4

**Summary:**

The paper presents a new training-free framework applicable to the verification of LLM outputs. The proposed approach called ReFeri re-uses the few shot examples given to the LLM to verify the output. It basically combines the score of the output sequence conditioned on the input examples with the score of the few shot examples conditioned on the output sequence. This is inspired by the classical Bayes rule. The experimental evaluation is carried out on several benchmark datasets including QA tasks as well as reasoning tasks, using GPT and Llama models. The results demonstrate improved performance compared with existing state-of-the-art approaches.

**Strengths:**

- Test-time scaling and verification of LLM outputs are crucial to improve LLM performance in complex tasks such as reasoning. Therefore, advances in LLM output verification are definitely called for.

- The proposed method is very efficient in the sense that it does not require additional training (as it is usually the case for RPMs).

**Weaknesses:**

- While the forward confidence score is fairly straightforward, I found the presentation of the backward confidence score a bit confusing. Perhaps an illustrative example would help to better convey the technical details.

- The performance measures reported in Table 1 lack standard deviations.

- The novelty of the proposed approach is fairly limited, it is basically a simple combination of sequence probabilities that seem to work in practice on a selection of datasets. The paper does not provide any theoretical justification as to why the proposed scheme works.

**Questions:**

- Aș far as I understand the proposed scheme requires token probabilities. If that's not the case then how do you compute the forward and backward consistency scores?

---

> ### Author Response · Authors · 2025-11-22
> **Response to Reviewer htr2 [1/2]**
>
> Dear Reviewer htr2,
>
> We appreciate your review with thoughtful comments. We have carefully considered each of your questions and provide detailed responses below.
>
> ---
>
> **[W1] I found the presentation of the backward confidence score a bit confusing. Perhaps an illustrative example would help to better convey the technical details.**
>
> Thank you for pointing out the presentation of the backward score. Regarding the illustrative example, we first note the qualitative token-level analysis presented in Appendix D.1. In that section, we visualize which tokens in the few-shot examples contribute most to the backward score. These analyses show that the backward term selectively attends to task-relevant components; for example, numerical symbols or answer-format tokens in MATH500 and GPQA.
>
> Next, to further provide insight about how the backward term works, we conducted an additional experiment designed to test whether a candidate with a high backward score indeed captures a task-level “rule” or “archetype.” Specifically, we first took the generated outputs from LLaMA, ranked them by their backward score, and used each output as a one-shot demonstration to solve the original few-shot queries. For a more intuitive understanding, we used the no-replace backward score, which is a simplified variant where each few-shot example $\mathbf{x_i}$ is evaluated in a one-shot manner using test query $\tilde q$ and the candidate response $r_k$ as condition:
>
> $S^\prime_{\texttt{Back}}(r_k) := \log P(a_{i^\dagger} \vert q_{i^\dagger}, \tilde{q}, r_k) - \log P(a_{i^\dagger} \vert q_{i^\dagger})$
>
> By utilizing the no-replace term, we isolate the specific impact of the candidate response $r_k$ on measuring a single few-shot example $x_i$, thereby eliminating the confounding factor of other few-shot demonstrations.
>
> \begin{array}{l|ccccc} \hline
> \text{Task}
> & 1 & 2 & 3 & 4 & 5 \newline \hline
> \text{MATH500}
> & 87.0 & 85.4 & 85.2 & 84.2 & 84.5 \newline
> \text{MuSR-ta}
> & 37.2 & 37.6 & 33.7 & 30.3 & 29.9 \newline \hline
> \end{array}
>
> As shown in the table above, we observe a clear correlation: responses with higher backward scores more effectively guide the model to answer the original few-shot examples. This indicates that the backward term is not merely a heuristic signal, but it reflects whether the candidate serves as a strong archetype for the underlying reasoning pattern present in the demonstrations.
>
> In other words, a high backward score suggests that the candidate not only aligns with the task distribution but also encapsulates the reasoning structure that generalizes across the few-shot examples. We will add this discussion to the revision for clarity in Appendix B.9.
>
> ---
>
> **[W2] The performance measures reported in Table 1 lack standard deviations.**
>
> We would like to clarify several important points regarding standard deviations:
>
> - First, the baseline methods (e.g., USC, CoT-WP), as well as ReFeri, all operate deterministically on the same fixed set of candidate responses. Therefore, once the candidate set is fixed, there is no source of randomness in the scoring process, and the reported results inherently have no variance. This is why standard deviations were not originally included in the main table.
>
> - Second, the variability arising from the stochastic generation process has already been evaluated through multiple independent runs in Table 9 of the original draft. These results measure variability across independently sampled candidate sets and compare performance against CoT-WP.
>
> Nevertheless, to directly address the reviewer’s concern, we repeated all experiments in Table 1 for two additional runs, yielding three independent trials in total. The updated Table 11 in the revision now includes the complete results along with standard deviations. Across all cases, ReFeri consistently outperforms the baselines (e.g., CoT-WP, USC). For example, with GPT-4o-mini, ReFeri achieves an average accuracy of 63.9 ± 0.5, outperforming USC (62.3 ± 0.5). Similarly, for GPT-4o, ReFeri reaches 71.5 ± 0.4, exceeding the newly added Self-Certainty baseline (69.7 ± 0.4).
>
> Overall, these results confirm that the improvements achieved by ReFeri are statistically reliable.

---

> ### Author Response · Authors · 2025-11-22
> **Response to Reviewer htr2 [2/2]**
>
> **[W3] The novelty of the proposed approach is fairly limited, and the paper does not provide theoretical justification.**
>
> With regard to novelty and the underlying theoretical motivation, we would like to clarify that ReFeri is NOT simply combining existing sequence probabilities. It was clearly motivated by our central research question: **how can few-shot examples be leveraged not only for generation but also for verification?** We believe that few-shot examples remain one of the most practical ways to inject human prior knowledge into LLMs, particularly for unfamiliar tasks where the model’s internal knowledge is insufficient. However, conventional few-shot prompting uses these examples only as input context during generation, relying entirely on the model’s intrinsic in-context learning behavior without any explicit mechanism for verifying candidate responses based on them.
>
> Motivated by this limitation, we introduce a **novel decomposition of the selection objective using Bayes’ rule (Eq. 5)**, which yields two complementary terms: a forward confidence score and a backward consistency score, both conditioned on few-shot examples. This derivation provides a clear theoretical justification for the structure of our method, rather than being a heuristic combination of log-probabilities.
>
> Intuitively, the two terms play fundamentally different roles:
>
> - The **forward score** evaluates how likely the candidate is to be generated given the few-shot context.
> - The **backward score**, which is novel to our work, evaluates how well the candidate aligns with the patterns, constraints, or implicit rules encoded in the few-shot examples.
>
> While the idea of forward likelihood has been partially explored in prior work (e.g., CoT-WP considers a variant focusing on the answer portion), the backward score is entirely new and arises naturally from our Bayes-motivated derivation. This term is crucial because it introduces an explicit verification mechanism grounded in few-shot demonstrations, a capability that is absent in prior training-free reranking methods.
>
> Thus, ReFeri is not merely a simple combination of probabilities; it introduces a new, theoretically grounded framework for reusing few-shot examples in a dual role, generation and verification, which, to the best of our knowledge, has not been explored in previous work.
>
> ---
>
> **[Q1] How do you compute the forward and backward consistency scores?**
>
> In principle, both forward and backward consistency scores are calculated from token-level log-probabilities. ReFeri, however, does not require these probabilities from the models that generate the responses. Instead, as denoted in line 141, we separate the generation model from the estimation model. That is, the generation models are used purely to produce candidate answers, while scoring is performed using a separate estimation model (LLaMA-3.1-8B-Instruct), as described in lines 263–264.
>
> This separation is deliberate and provides two practical benefits:
>
> - First, ReFeri can be applied directly to closed-source or API-based LLMs (e.g., GPT-4o-mini and GPT-4o in Table 1) that do not expose token probabilities, because it only requires text outputs from the generator.
> - Second, since the estimation model is modular and independent, it can be chosen to be much smaller than the generator, reducing overall computational cost. As shown in Figure 4 and Table 16, ReFei maintains strong performance even with lightweight estimators, demonstrating that accurate scoring does not require a generator-sized model.
>
> We believe that this effectiveness, even when the generation and estimation models are not identical, is a key strength of our approach. This design choice makes ReFeri broadly applicable and can be easily integrated to existing pipelines.
>
> ---
>
> If you have any questions/concerns, please let us know.
>
> Thank you very much,
> Authors

---

### Official Review · Reviewer_D9LY · 2025-11-01

**Soundness:** 4
**Presentation:** 4
**Contribution:** 2
**Rating:** 6
**Confidence:** 3

**Summary:**

This paper introduces Referi (Recycle Few-Shot Examples to Verify LLM Outputs), a training-free framework for best-of-N selection in large language models. Current state-of-the-art approaches rely on output or process reward models, which require additional training for each target task. In contrast, Referi leverages few-shot examples and uncertainty estimation to perform selection without any task-specific training. It computes two complementary signals: (1) a forward confidence score, which measures the model’s confidence in its generated output given the few-shot examples, and (2) a backward confidence score, which assesses the confidence of the few-shot examples given the generated output. The framework is grounded in a Bayesian formulation, and its effectiveness is supported by experimental results.

**Strengths:**

- Paper is well-written and easy to follow.
- The authors motivate their proposal well and support the effectiveness of their method through various additional experiments.
- Even though the confidence score calculation and usage already exist in the literature and do not provide much to the community, I liked the incorporation of the backward confidence score.

**Weaknesses:**

- A recent related work, which also uses uncertainty for best of N selection,  is missing in the evaluations and related works:

 Zhewei Kang et al, Scalable Best-of-N Selection for Large Language Models via Self-Certainty, 2025

- I think one simple and crucial baseline is missing: self-consistency (majority voting). Selecting the most common answer across sampled generations. This is a stronger baseline than random selection.

- Seeing the golden performance would be helpful for better analysis. I.e, the performance of a golden selection mechanism (always selecting the correct option if it exists).

- There is an implicit assumption that the probabilities of the estimation model are well-calibrated. However, this is not the case in the real world. How would the calibration level of the estimation model affect Referi? I think this situation might amplify bias in certain cases, which should be added as a discussion in the paper.

**Questions:**

See weaknesses.

---

> ### Author Response · Authors · 2025-11-22
> **Response to Reviewer D9LY [1/3]**
>
> Dear Reviewer D9LY,
>
> We appreciate your review with thoughtful comments. We have carefully considered each of your questions and provide detailed responses below.
>
> ---
>
> **[W1] Comparison with “Self-Certainty” (Kang et al)**
>
> We appreciate the reviewer for pointing out this missing relevant work. Self-Certainty [1] indeed estimates the uncertainty of responses using the predictive distribution, and thus can be viewed as a purely forward-only method, conceptually similar to CoT-WP, which is already included as a key baseline in our evaluations.
>
> Nevertheless, we agree that an empirical comparison with Self-Certainty would further strengthen our study. We therefore conducted additional experiments, and the results are presented in the updated Table 11. While Self-Certainty is a strong and competitive baseline, ReFeri consistently outperforms. For example, on the GPT-4o-mini model, ReFeri achieves an average accuracy of $63.9 \pm 0.5$, exceeding the $62.2 \pm 0.2$ compared to the Self-Certainty baseline. Similarly, on the GPT-4o, ReFeri also outperformed on $71.5 \pm 0.4$ compared to $69.7 \pm 0.4$. For a fair comparison, and to remain consistent with our evaluation setup, we used LLaMA-3.1-8B-Instruct to compute the self-certainty scores, since log probabilities for closed-source models such as GPT-4o are not directly accessible.
>
> Interestingly, while Self-Certainty provides notable benefits for smaller open-source models (e.g., LLaMA), which often produce more verbose or stylistically variable responses, its effectiveness diminishes for stronger models such as GPT-4o-mini and GPT-4o. We conjecture this is because the sampled responses In these large models are uniformly high-quality and have very similar predictive distributions, making entropy-based uncertainty insufficient for distinguishing subtle differences.
>
> In contrast, ReFeri remains effective because it leverages few-shot demonstrations during validation, not just generation. Few-shot examples offer a compact way to inject human prior knowledge, and ReFeri uses this external information to complement internal model confidence. As a result, unlike forward-only approaches that depend solely on the model’s intrinsic distributional signals, ReFeri incorporates external human insights and generalizes more reliably across models and tasks.
>
> These observations suggest that while forward-only metrics become unreliable in realistic scenarios where powerful LLMs produce uniformly confident outputs, ReFeri maintains robustness by integrating complementary backward information grounded in few-shot demonstrations. We have added these results and corresponding discussion in Appendix B.5.
>
> ---
>
> **[W2] Simple and crucial baseline is missing: self-consistency (majority voting)**
>
> As noted in lines 40–41, self-consistency has an inherent limitation: it is only applicable when answers can be easily extracted and normalized for voting. For example, it is hard to be applied for open-ended text generation. For this reason, rather than the original self-consistency, we include Universal Self-Consistency (USC) as a baseline, which uses prompt-based evaluation to select among free-form outputs. In fact, HotpotQA is a representative case where standard self-consistency cannot be reliably applied, since mapping free-form outputs into consistent discrete answer categories is non-trivial.
>
> Nevertheless, to address the reviewer’s concern, we conducted additional evaluations using standard self-consistency. To enable it on HotpotQA, we identified overlapping shared spans among free-form responses, counted how many responses contain each span, and used these aggregated counts as voting scores. In addition, inspired by the weighted voting scheme in Self-Certainty [1], we experimented with a Borda-voting method that uses the ReFeri metric as the weight, fixing the parameter p=1 to avoid introducing new hyperparameters.
> Let $\text{rank}(k)$ denote the rank of candidate $r_k$ based on its ReFeri score.
> The corresponding borda weight is calculated as follows:
>
> $$w_k = (K - \text{rank}(k) + 1)^p$$
>
> The final selection score of candidate k is obtained by summing the weights of all candidates who make the same prediction:
>
> $$k^\star_{\text{Borda}}= \arg\max_{k \in \{1,\dots,K\}}\sum_{j=1}^{K}  w_k*\mathbf{1}[r_j = r_k]$$
> The results are presented in the table below. Overall, ReFeri continues to outperform self-consistency on average. For instance, on MuSR-ta with LLaMA, ReFeri shows substantially higher performance (e.g., 76.4 vs. 79.6). More importantly, these methods are complementary: applying Borda voting with ReFeri yields notable improvements over conventional self-consistency, particularly on LLaMA. We have added these results and the corresponding discussion in Appendix B.2.

---

> ### Author Response · Authors · 2025-11-22
> **Response to Reviewer D9LY [2/3]**
>
> **[W2] Simple and crucial baseline is missing: self-consistency (majority voting) (con't)**
>
> \begin{array}{c|c|ccccccc|c} \hline
> \text{Models} & \text{Methods} & \text{MuSR-ta} & \text{MuSR-op} & \text{GPQA} & \text{MATH500} & \text{DROP (EM/F1)} & \text{HotpotQA (EM/F1)} & \text{MMLU-Pro} & \text{Avg.} \newline \hline
> \text{LLaMA-3.1-8B-It} & \text{Self-Consistency} & 76.4 & 54.7 & 26.3 & 52 & 73.0/78.0 & 23.0/29.8 & 45.1 & 50.1 \newline
> & \text{ReFeri}& 79.6 & 57.8 & 35.4 & 51.2 & 69.4/75.7 & 25.0/33.2 & 45.1 & 51.9 \newline
> & \text{+ Borda-vote (p=1)}& 78.4 & 57.8 & 30.3 & 53.2 & 74.0/79.7 & 26.8/34.9 & 46.4 & 52.4 \newline \hline
> \text{GPT-4o-mini} & \text{Self-Consistency}& 84.8 & 60.6 & 43.4 & 79.6 & 80.0/85.3 & 35.6/46.8 & 65.3 & 64.2 \newline
> & \text{ReFeri}& 82.8 & 61.3 & 41.9 & 77.8 & 79.2/84.9 & 36.2/48.0 & 64.9 & 63.4 \newline
> & \text{+ Borda-vote (p=1)}& 86.8 & 60.9 & 44.4 & 79.4 & 80.4/86.0 & 36.8/48.3 & 65.7 & 64.9 \newline \hline
> \text{GPT-4o} & \text{Self-Consistency}& 86.8 & 71.5 & 50.5 & 80 & 84.4/91.6 & 45.6/60.5 & 76.4 & 70.7 \newline
> & \text{ReFeri}& 90.4 & 71.9 & 51.5 & 77.8 & 83.6/91.1 & 47.0/60.7 & 75.4 & 71.1 \newline
> & \text{+ Borda-vote (p=1)}& 88.8 & 71.5 & 49.5 & 80.2 & 84.8/92.0 & 48.0/61.9 & 76.2 & 71.3 \newline \hline
> \end{array}
>
> On the other hand, to further demonstrate the practical value of our method independent of response normalization more clearly, we also evaluated ReFeri on open-ended personalization tasks where self-consistency evaluation cannot be applied at all. To this end, we consider LaMP benchmark [2] for evaluating LLM personalization; LaMP-4 and LaMP-5 are structurally closer to summarization than QA, and their evaluation metrics rely on ROUGE rather than answer accuracy. The results are presented below:
>
> \begin{array}{l|cc|cc} \hline
> \text{Methods}
> &  {\text{LaMP-4}}  &  &{\text{LaMP-5}} \newline
> & \text{Rouge-1} & \text{Rouge-L} & \text{Rouge-1} & \text{Rouge-L} \newline \hline
> \text{Vanilla}        & 0.120 & 0.106 & 0.421 & 0.332 \newline
> \text{Few-shot RAG}   & 0.138 & 0.123 & 0.451 & 0.366 \newline \hline
> \text{Ours}           & \mathbf{0.143} & \mathbf{0.128}  & \mathbf{0.470} & \mathbf{0.394} \newline \hline
> \end{array}
>
> Here, ReFeri consistently outperforms both Vanilla and the Few-shot RAG (N=3) baseline on both ROUGE-1 and ROUGE-L. These results demonstrate that ReFeri can be naturally extended to open-ended generation tasks, unlike self-consistency, which fundamentally relies on answer parsing. We have added these results and the corresponding discussion in Appendix B.3.
>
> ---
>
> **[W3] Seeing the golden performance**
>
> We thank the reviewer for suggesting the inclusion of the oracle upper bound. We have added the Oracle (Golden) performance to Appendix B.1 (Table 5) as requested.

---

> ### Author Response · Authors · 2025-11-22
> **Response to Reviewer D9LY [3/3]**
>
> **[W4] How would the calibration level of the estimation model affect Referi?**
>
> Thank you for your insightful question regarding the calibration of the estimation model. To investigate this, we conducted a series of experiments that explicitly altered model calibration via temperature scaling, adjusting the temperature applied to the estimation model’s logits to T = 0.5, 1.0 (default), 1.5, and 2.0. As shown in the table below, ReFeri remains remarkably robust across temperatures, exhibiting only minimal performance variation even under strong overconfidence or underconfidence. We include these results and the corresponding discussion in Appendix B.10.
>
> \begin{array}{c|c|ccccc}
> \hline
> \text{Models} & \text{Temp} & \text{MuSR-ta} & \text{MuSR-op} & \text{GPQA} & \text{MATH500} & \text{DROP (EM/F1)} & \text{HotpotQA (EM/F1)}  & \text{Avg.} \newline \hline
> \text{LLaMA-3.1-8B} & 0.5 & 79.6 & 57  & 33.3 & 50.6 & 69.8 /75.9  & 24.6 /33.0  & 52.5 &  \newline
> & 1   & 79.6 & 57.8 & 35.4 & 51.2 & 69.4 / 75.7 & 25.0 / 33.2 & 53.1 & \newline
> & 1.5 & 78.8 & 57   & 34.8 & 52   & 70.2 /76.1  & 25.2 /33.2  & 53.0 &  \newline
> & 2   & 78   & 56.2 & 34.3 & 51.8 & 71.0 /77.1  & 24.6 /32.7  & 52.7 &  \newline\hline
> \text{GPT-4o-mini} & 0.5 & 82   & 59.4 & 43.9 & 78   & 78.0 /83.8  & 35.8 /47.1  & 62.9 &  \newline
> & 1   & 82.8 & 61.3 & 41.9 & 77.8 & 79.2 / 84.9 & 36.2 / 48.0 & 63.2 &  \newline
> & 1.5 & 83.6 & 60.2 & 43.4 & 78   & 78.0 /84.7  & 36.4 /48.3  & 63.3 &  \newline
> & 2   & 83.2 & 59   & 41.4 & 78.4 & 77.4 /84.1  & 35.2 /47.6  & 62.4 &  \newline\hline
> \text{GPT-4o} & 0.5 & 90.4 & 71.1 & 52   & 77.4 & 84.0 /91.3  & 47.6 /60.7  & 70.4 &  \newline
> & 1   & 90.4 & 71.9 & 51.5 & 77.8 & 83.6 / 91.1 & 47.0 / 60.7 & 70.4 &  \newline
> & 1.5 & 91.2 & 71.1 & 50.5 & 77.6 & 83.4 /91.1  & 45.0 /59.2  & 69.8 &  \newline
> & 2   & 90.4 & 71.9 & 51   & 77.6 & 81.8 /89.9  & 44.8 /59.3  & 69.6 &  \newline\hline
> \end{array}
>
> We note that this robustness is further supported by our cross-model and cross-scale evaluations. As shown in Figure 4 and Appendix B.8 (Table 17), ReFeri with a 1B-scale estimation model maintains stable performance across diverse tasks and calibration settings, whereas the likelihood-based baseline CoT-WP exhibits significantly greater variability.
>
> Overall, these observations suggest that ReFeri does not heavily rely on precise calibration of the estimation model. The backward term provides an additional alignment signal, capturing how well a candidate response explains the few-shot examples, which compensate for calibration drift by capturing explanatory alignment with the few-shot examples rather than relying solely on raw likelihood peaks.
>
> ---
>
> [1]  Kang et al, Scalable Best-of-N Selection for Large Language Models via Self-Certainty, 2025
> [2] Salemi et al., Lamp: When large language models meet personalization. ACL2024
>
> ---
>
> If you have any questions/concerns, please let us know.
>
> Thank you very much,
> Authors

---

### Author Response · Authors · 2025-11-22
**General Response**

Dear reviewers and AC,

We appreciate you for taking the time and making the effort to examine our manuscript.
As the reviewers highlighted, we propose a **well-motivated method (D9LY)** that offers a **novel perspective on recycling few-shot examples (UCFk)** by incorporating a backward score, while being **efficient, training-free (htr2), and broadly generalizable (UCFk)**. Our approach is supported by **comprehensive experiments (D9LY, UCFk) with clear writing (D9LY)**.

We appreciate your constructive feedback on our manuscript. In response, we have meticulously updated and enhanced the manuscript, including the following additional discussions and experiments:

- Clearer explanation about the estimation model (Section 2, Section 3.1 Implementation details)
- Additional baselines Self-Certainty and Self-Consistency (Appendix B.2, B.5, Tables 8 and 11)
- Gold-label performance for reference (Appendix B.2, Table 5)
- Effects of model calibration (Appendix B.10, Table 20)
- Lack of standard deviation (Appendix B.5, Table 11)
- More details about the backward score (Appendix B.9, Table19)
- The theoretical motivation regarding "lightweight approximation" (Appendix B.8)
- More challenging open-ended task (Appendix B.3, Table 9)

For your convenience, revisions have been temporarily marked in $\textbf{\color{green}green}$ in the amended manuscript. We sincerely believe that these updates may help us better deliver the benefits of the proposed ReFeri to the ICLR community.

Thank you very much,
Authors.

---

### Author Response · Authors · 2025-11-26
**A Gentle Invitation for Further Discussion**

Dear reviewers,

Once again, we would like to kindly express our appreciation for your thoughtful reviews and the time you have devoted to evaluating our submission.

If you have any additional thoughts or follow-up comments after reading our author response, we would be very grateful to hear them. Your insights during the discussion phase are extremely valuable and help us further clarify and strengthen the work.

Thank you very much for your time and engagement.

Kindest regards,
Authors

---

### Meta-Review · Area_Chair_v3tF · 2026-01-02

**Summary:**

This paper proposes ReFeri, a training-free verification and selection framework for large language models that reuses few-shot demonstrations to score candidate answers. While the paper is clearly written and addresses a relevant practical problem, the core contribution does not rise to the level expected for ICLR as argued below.

(1) Incremental novelty and weak conceptual contribution. The central idea, scoring candidate answers using log-likelihoods conditioned on prompts or demonstrations, is quite relevant to the prior work (e.g., self-consistency, self-certainty, likelihood-based reranking, LLM-as-a-judge variants). Although the authors frame this via a Bayes-rule decomposition, this framing is a bit post-hoc and does not yield sufficient new theoretical insights or guarantees. The rebuttal clarifies the intuition but does not fundamentally elevate the contribution beyond a heuristic extension.

(2) Empirical gains are modest. The reported improvements over baselines are generally small, and while the authors added standard deviations and additional baselines in the rebuttal, the gains remain incremental rather than compelling. In several settings, performance is also comparable to simpler methods such as self-consistency, raising questions about whether the added complexity is justified.

(3) Cost/complexity tradeoffs. While the authors provide runtime comparisons and propose lightweight approximations, the method’s practical attractiveness remains highly dependent on several factors. As a result, although the approach is training-free, it is not compute-free and can incur substantial inference-time overhead. This questions the practicality of the proposed approach.

In light of the above concerns, I recommend rejection despite the authors’ effortful rebuttal.

**Reviewer Concerns:**

+ The authors made a great effort to address several presentation- and evaluation-level concerns. In particular, they added missing baselines, reported standard deviations across multiple runs, clarified the role of the backward score with additional analysis, and provided further discussion on calibration and runtime comparisons.

- The core concerns regarding incremental novelty remain unresolved.

**Reviewer Scores:**

Reviewer D9LY may keep the score 6.
Reviewer htr2 may keep the score 4.
Reviewer UCFk may raise the score to 4.

---

### Decision · Program_Chairs · 2026-01-26

Reject